# Cold adaptation recorded in tree rings highlights risks associated with climate change and assisted migration

David Montwé[1], Miriam Isaac-Renton[1], Andreas Hamann [1] & Heinrich Spiecker [2]

With lengthening growing seasons but increased temperature variability under climate change, frost damage to plants may remain a risk and could be exacerbated by poleward planting of warm-adapted seed sources. Here, we study cold adaptation of tree populations in a wide-ranging coniferous species in western North America to inform limits to seed transfer. Using tree-ring signatures of cold damage from common garden trials designed to study genetic population differentiation, we find opposing geographic clines for spring frost and fall frost damage. Provenances from northern regions are sensitive to spring frosts, while the more productive provenances from central and southern regions are more susceptible to fall frosts. Transferring the southern, warm-adapted genotypes northward causes a significant loss of growth and a permanent rank change after a spring frost event. We conclude that cold adaptation should remain an important consideration when implementing seed transfers designed to mitigate harmful effects of climate change.

[1] Department of Renewable Resources, University of Alberta, 751 General Services Building, Edmonton, AB T6G 2H1, Canada. [2] Chair of Forest Growth and Dendroecology, Albert-Ludwigs-Universität Freiburg, Freiburg, Germany 79106. These authors contributed equally: David Montwé, Miriam Isaac-Renton. Correspondence and requests for materials should be addressed to D.Mé. (email: david.montwe@ualberta.ca)

The pace of observed climate warming, particularly in northern latitudes, implies that environmental conditions are changing faster than plant populations can adapt, acclimate or migrate[1–3]. This mis-match is expected to destabilize ecosystems by decreasing productivity while increasing mortality and susceptibility to insect attacks and diseases[4–8]. Maladapted plant populations may further contribute to climate change through reduced carbon sequestration, or may even turn into significant sources of carbon dioxide[9]. To mitigate such consequences, assisted migration has been proposed as part of regular reforestation programs[10,11] and as part of conservation efforts[12]. Assisted migration involves changing the genetic composition of plant populations by moving seed material to climate regions where they are anticipated to be well adapted in the future[2,3,10,11]. For example, transferring seeds from southern, drought-tolerant tree populations may help more northern forests adapt to drier, warmer conditions[2,3,13,14]. However, there are concerns that assisted migration may not be successful and lead to unintended consequences[15,16]. For example, planting warm-adapted populations in anticipation of a warmer future could expose these seedlings to frost damage and compromise their survival[17–19]. As climates warm and changing phenology interacts with increasing temperature variability[20–25], frost damage will remain a risk in the future. Understanding the adaptation of plant populations to cold—as well as to heat and drought—is critical for minimizing risks under changing climates and assisted migration[26].

Due to their economic importance, adaptation and maladaptation of trees have been studied intensively with genetic provenance trial series, also known as common garden experiments[13,27]. Seeds collected from across a species range (i.e., provenances are the geographic locations from which seed sources originate) are grown at multiple planting sites to reveal intra-specific genetic differentiation. Such reciprocal transplant experiments are ideal for studying plant–climate interactions and for assessing the risks involved in seed transfer to new locations[28–30]. Growing northern seed sources at southern planting sites involves a climate transfer that can simulate projected warming, with the growth response being predictive of future performance under realistic long-term field conditions. Conversely, moving southern seeds northward can test the performance of warm-adapted populations under currently colder environments, thereby suggesting limits to seed transfer. Within these designs, genetic adaptation to cold in tree populations has been studied by observation of tissue damage after sporadic natural frost events and by exposing collected tissue samples to artificial freezing tests in the laboratory[17,31–34].

Tree rings can provide additional information for evaluating cold adaptation since they provide a record of how past climate has influenced growth. Two wood anatomical features have been linked to cold and frost. First, layers of deformed, collapsed tracheids and traumatic parenchyma cells have been described as frost rings[19,35,36]. Frost rings have been linked to generally colder years[37] in which air temperatures fall below freezing during a period of cambial activity[19]. Second, light rings consist of layers of cells that are incompletely lignified[38] and have been associated with frost events that kill the cells before the lignification process is complete[39,40]. Due to the lack of lignin, light rings do not fully absorb the red Safranin dye during a double staining procedure with Astrablue. This staining procedure causes non-lignified layers of cells to appear blue, hence light rings have also been referred to as blue rings[41]. These blue rings (failed lignification)[41] and frost rings (cambium damage)[19,35,37] therefore act as long-term archives of cold damage.

Here, we study these cold and frost signatures in trees grown in provenance trials, allowing us to quantify genetic differentiation in susceptibility to cold among populations. This can inform limits to seed transfers designed to address projected climate change. Our study species is lodgepole pine (Pinus contorta Dougl. ex. Loud.), represented by a range-wide collection of seed sources grown for three decades in the Illingworth provenance trial[42]. A subset of 20 provenances from across western Canada and the United States, grown at three central planting sites, were selected to analyze the response to climate transfers. We first investigate whether frost imprints in tree rings can act as a reliable indicator of cold susceptibility by analyzing the association of climate and cold events at the test sites with the occurrence of blue and frost rings. We further analyze frost imprints in the context of long-term growth data to quantify potential impacts of cold damage on productivity.

Provenances from the northern extent of the species range were resistant to fall cold events but susceptible to spring frosts, implying a maladaptation under projected climate warming. Central populations were most productive but also most sensitive to early fall frosts. Meanwhile, transferring the most southern, warm-adapted genotypes northward caused rank-changing growth loss after a spring frost event. We conclude that opposing clines in spring and fall cold adaptation represent risks of both climate change and assisted migration.

## Results

**Annual climate variation and intensity of cold damage.** In total, we examined 2999 tree rings from 117 trees from 20 different provenances grouped into four regional populations. Across all provenances, we recorded the occurrence of 689 blue rings (23%) and 204 frost rings (7%). In addition to occurrence, we assigned intensity scores, ranging from 1 to 5 (0 indicates absence). Blue rings occurred exclusively in the latewood (Fig. 1a), while frost ring damage occurred most often in the first rows of cells in the earlywood, labeled as position 1. The occurrence of the most severe blue rings in fall was usually followed by frost rings at position 1 (Fig. 1a). It therefore appears that the blue ring captures a cold event at the end of the growing season that leads to cambial damage and irregular growth upon re-activation in the following spring. Occasionally, frost rings occurred in the earlywood after some tracheids had formed normally (Fig. 1b). These were labeled as frost rings in position 2 and we interpret them as damage caused by late spring frost events. Higher occurrences of blue and frost rings in the first ten years of the experiment (Fig. 1d) may reflect higher vulnerability of small saplings to frost (Supplementary Fig 2). Increasing diameter decreased the odds of observing more severe blue ring scores ($\chi^2 = 550.6$, $p < 0.001$, Supplementary Table 4), frost rings at position 1 ($\chi^2 = 398.9$, $p < 0.001$, Supplementary Table 6), and position 2 ($\chi^2 = 54.8$, $p < 0.001$, Supplementary Table 8). This may be due to their smaller size and lower heat absorption, more tender growth from second flushing in younger trees, and their proximity to the soil surface, which is colder due to radiative cooling[36,43–46].

An analysis of climate data at the common garden sites reveals that blue rings and frost rings in position 1 are linked to a late initiation of the growing season, an early end of the growing season, and a generally cool growing season (Supplementary Tables 3 and 5, also apparent by comparing Fig. 1d, e). Because lignification can lag behind cellular development by several weeks[37,38,47–49], the growing season length and overall growing conditions during the season have an impact on the occurrence of frost and blue rings if the lignification process is incomplete by the time the first fall frosts occur. For example, the year 1999 had high blue ring intensities, and was associated with both a cool spring and summer (Fig. 1f) and a low accumulation of warmth measured in growing degree days (Fig. 1g). In contrast, no frost imprint was observed in 1998. This year had a long growing

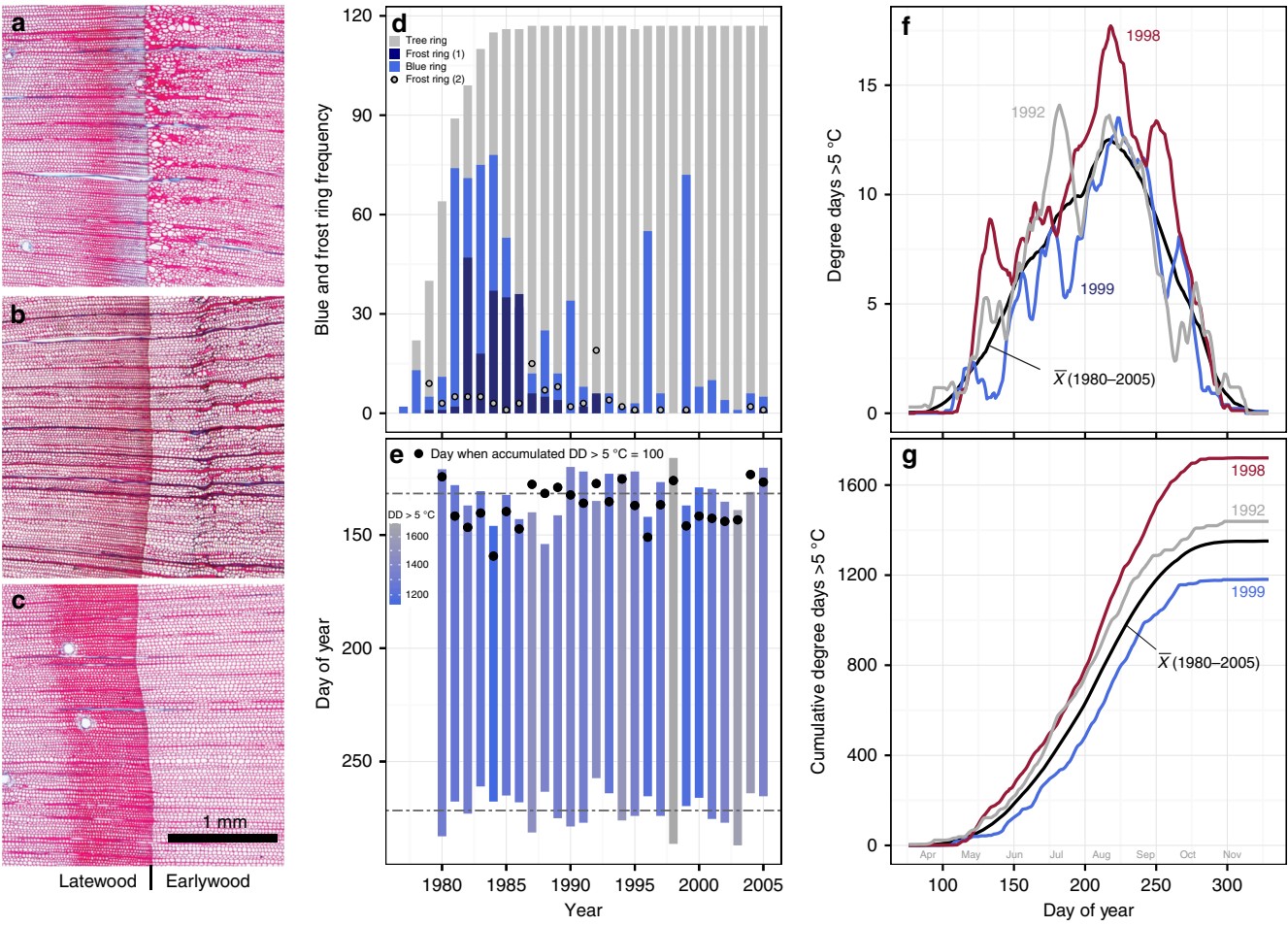

**Fig. 1** Blue and frost rings as a function of growing degree days. Stained micro sections of tree rings are shown in **a–c**. Safranin dyes lignin red and Astrablue dyes cellulose blue, so that the dual-staining procedure results in a blue ring where a layer of cells lacks lignin. **a** An example for a blue ring of high intensity with a subsequent frost ring in position 1 (i.e., first cells of earlywood indicative of fall frost damage). **b** A frost ring in position 2 (i.e., later in the earlywood indicative of spring frost damage), while **c** shows a normal tree ring. **d** The occurrence of blue and frost rings by year. Gray bars indicate the sample size per year (total $n = 2999$), light blue bars indicate the occurrence of blue rings, dark blue bars represent frost rings occurring in position 1, and the gray points represent frost rings in position 2. **e** Bars represent growing season defined as frost free days, black circles represent early warming in spring, as the date on which cumulative degree days above 5 °C exceeded 100. Horizontal dashed lines indicate the average start and end of the growing seasons. **f** Examples of growing degree days above 5 °C among for a year with high occurrence of blue rings (1999), a year with frost rings in position 2 (1992), a year with no blue or frost rings (1998), as well as the overall average (1980–2005). The lines are smoothed using a 15-day running mean. **g** The cumulative effect of the growing degree signatures shown in **f**

season (Fig. 1e), warm spring, hot summer and warm fall (Fig. 1f), which led to an above-average annual accumulation of growing degree days above 5 °C (Fig. 1g).

In contrast, frost rings in position 2 reflect cold events in spring after growth has been initiated. Here, we found the strongest climate relationships to be with spring temperature accumulation and the start of the growing season (Supplementary Table 7). As an example, the year 1992 had a high proportion of frost rings in position 2, and a false spring: an unusually early warm spell in April before day 100 (Fig. 1f).

**Genetic differences in susceptibility to cold damage.** Genetic differentiation of the four regional lodgepole pine populations was statistically significant for blue ring intensity ($\chi^2 = 46.7$, $p < 0.001$, Supplementary Table 3) and frost ring intensity at position 1, presumed to be fall frost damage ($\chi^2 = 12.9$, $p = 0.005$, Supplementary Table 5). See Table 1 for multiple comparisons among regional populations and the three cold response variables. The Southern Interior provenances were most affected by blue rings

| Table 1 Multiple comparisons among populations' blue and frost ring intensity scores | | | |
|---|---|---|---|
| | **Blue ring score** | **Frost ring score** | |
| | | **Position 1** | **Position 2** |
| N – CI == 0 | **<0.001** | 0.997 | 0.986 |
| SI – CI == 0 | **0.003** | **0.006** | 0.448 |
| US – CI == 0 | 0.163 | 0.403 | 0.068 |
| SI – N == 0 | **<0.001** | **0.003** | 0.278 |
| US – N == 0 | **<0.001** | 0.298 | **0.034** |
| US – SI == 0 | 0.506 | 0.335 | 0.690 |

*P*-values were adjusted for multiple comparisons with the Tukey method. Bold indicates significance at the 0.05 threshold

and fall frost rings, followed by the United States (US) group. The Northern provenances had the lowest levels of blue rings and fall frost rings, with the Central Interior provenances showed intermediate susceptibility (Fig. 2a, b). In contrast, Northern

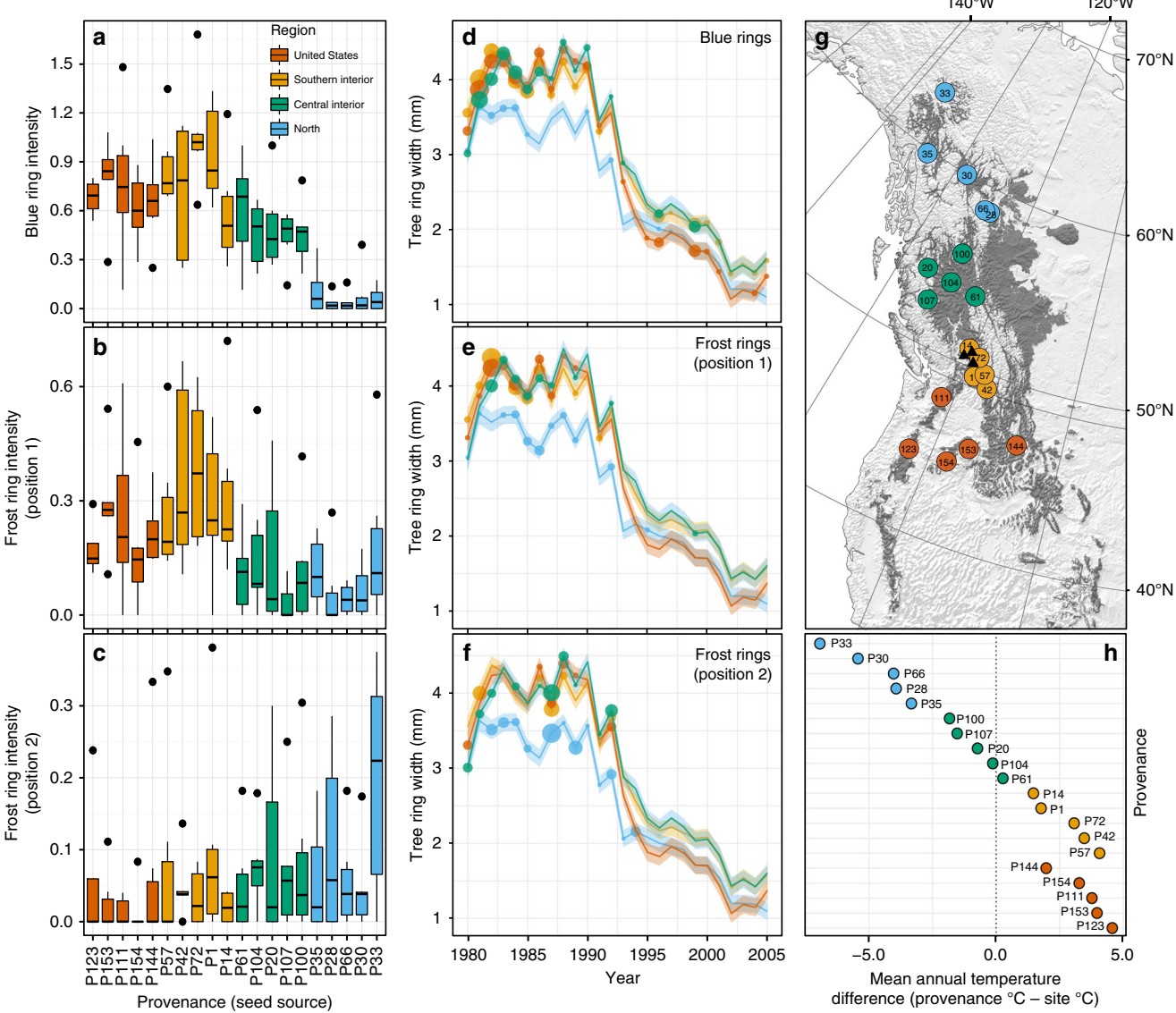

**Fig. 2** Blue and frost ring intensities by population and associated annual growth. **a–c** The distribution and medians of blue ring and frost ring intensities, where provenances are sorted by the mean annual temperature of their source climate (warmest on the left, n = 117). **d–f** Show how growth is related to blue and frost ring intensities on an annual basis, where average blue and frost ring intensities (including zero values) are represented by the size of the circles. Provenances are colored according to their region and labeled by their identification number. **g** The location of provenances and regions as well as the species range of lodgepole pine (dark gray) and the location of test sites sampled in this study (black triangles). The map was produced by the authors with ArcInfo 10.1 using vector and raster data from www.naturalearthdata.com (Public Domain). The species range layer is based on presence/absence raster data created by the authors. **h** The difference of the provenance source climate to the average climate of the test sites thereby indicates the degree and direction of climate transfers

provenances were most susceptible to spring frost damage as indicated by frost rings in position 2 ($\chi^2 = 9.2$, $p = 0.026$, Supplementary Table 7). Notably, the geographic cline in vulnerability to spring frosts (Fig. 2c) appears opposite to the geographic cline for fall frosts (Fig. 2a, b), with the most northern and coldest location of origin showing the highest intensity of frost rings in position 2 (Fig. 2g, h). Similarly, correlations to climate of seed origin (Table 2) confirm that provenances adapted to northern, colder environments with larger temperature differences and shorter growing seasons incur less blue rings and fall frost damage but are more susceptible to spring frosts.

## Discussion

The data suggest that the northern population may be most vulnerable to late spring frost events and false springs under

climate change[23,50], especially if temperature patterns become more variable[22,51]. The likely explanation for this counter-intuitive observation is that the northern population is adapted to a short growing season. Northern provenances flush earlier than southern seed sources for a given chilling and heat sum (measured in degree days) to take advantage of a limited growing season[46]. However, this adaptive strategy may no longer be successful under anticipated climate change. This is especially relevant since temperature is more influential for spring flushing than photoperiod in most forest trees[52]. Moreover, the resulting frost damage can reduce growth and survival[19]. Such damage could also reduce wood quality and value by creating weaknesses and defects in the timber[53].

Another equally important finding is that this study confirms the need for transfer limits under assisted migration. Both the US

**Table 2 Correlation between cold damage and climate of provenance origin**

| Climate variable | Blue rings | Frost rings pos. 1 | Frost rings pos. 2 |
|---|---|---|---|
| Latitude | **−0.81** | **−0.58** | **0.54** |
| Longitude | **0.72** | **0.54** | **−0.58** |
| Elevation (m; above sea level) | 0.11 | 0.00 | −0.24 |
| Mean annual temperature (°C) | **0.89** | **0.70** | **−0.56** |
| Mean coldest month temperature (°C) | **0.83** | **0.58** | **−0.57** |
| Temperature difference (°C) | **−0.70** | −0.37 | **0.55** |
| Mean summer precipitation (mm) | −0.42 | −0.42 | 0.09 |
| Summer heat–moisture index | **0.48** | 0.45 | −0.14 |
| Annual sum of degree days >5 °C | **0.83** | **0.82** | −0.42 |
| Number of frost free days | **0.84** | **0.82** | **−0.52** |
| Growing season length (no. of days) | **0.76** | **0.83** | −0.47 |

Pearson's correlation coefficients of average blue and frost ring intensities and provenance climate (1961–1990 climate normal)
Bold values indicate significance after correcting p-values for the number of climate variables with the Benjamini–Hochberg method. The correlation of average blue ring and frost ring intensities are based on 20 values (climate conditions at the origin of 20 seed collection sites)

and Southern Interior provenances were transferred to colder environments in this experiment (Fig. 2h), with the test sites located either further north or higher in elevation than the provenance origins. Notably, the most productive Southern Interior provenances also showed the highest fall frost damage (Fig. 2d, e, Supplementary Fig 1a, b). Given that short growing seasons and low cumulative growing degree days are a likely cause of blue rings and frost rings in fall, a transfer of these seed sources to colder environments is likely to exacerbate this problem. While the US provenances did not incur additional fall cold damage relative to the Southern Interior region, spring frost susceptibility compromised their growth: A substantial decline in the US population's tree-ring growth began shortly after a spring frost event in 1992 (Fig. 2f). Prior to this rank change ($p < 0.001$, Gail–Simon Test), these more southern seed sources had shown equal productivity to the local seed sources. Regarding transfers from colder to warmer environments, we observed an expected cline towards minimal blue-ring occurrences in northern provenances (Fig. 2a), but no significant difference between the northern and central interior populations in fall frost damage (Fig. 2b).

We conclude that expanding allowable distances of seed transfers can increase the risk of frost damage and potentially reduce growth. Cold adaptation should remain an important consideration when implementing long-distance seed transfers in temperate ecosystems designed to mitigate harmful effects of climate change. The study revealed opposing geographic clines for spring and fall frost damage, which both need to be considered to devise effective strategies to minimize risks in assisted migration prescriptions to address climate change. Provenance trials have been established for most of the major tree species of the temperate and boreal zone, and tree-ring analysis can be conducted non-destructively with increment borers to take full advantage of the information contained in these valuable transplant experiments. In tree improvement programs, breeding populations could be screened for cold tolerance[31,54].

Although maladaptation under climate change is of concern for forests due to the long lifespan of trees, genetic maladaptation may occur in other widespread species where local adaptation exists. Assisted migration is being considered more broadly as a means of conserving biodiversity under climate change[55]. If the opposite clines in cold adaptation found here apply more generally, this would highlight the risk trade-offs associated with inaction versus those of assisted migration. In addition to diversification of seed sources[2,56], the most suitable course of action in forestry as well as in conservation may thus involve shorter seed transfer distances, equivalent to planning for a shorter climate change time-frame[56].

## Methods

**Experimental design and field methods.** Stem disks were cut at 1.3 m from trees of selected lodgepole pine (*Pinus contorta*) populations in the Illingworth provenance trial[42]. This experiment tested 153 seed sources, or provenances, at 60 planting sites in an incomplete randomized block design[42]. One-year-old seedlings were planted in 1974 and most trees died in 2006 due to the mountain pine beetle epidemic. We used a representative sub-set from this trial for detailed wood anatomical analyses: Five provenances from British Columbia's North and southern Yukon (N); five provenances each from British Columbia's Central Interior (CI) and Southern Interior (SI); and five provenances from the United States (US). These 20 provenances were replicated on three experimental sites in British Columbia's Southern Interior (Fig. 1; Supplementary Tables 1 and 2). Each site consists of two blocks, with provenances planted in plots of nine trees. From each plot, we selected one tree of median height for wood anatomical analyses, although three samples were not suitable due to early mortality or rot. Thus, a subsample of 117 trees was used.

**Preparation of micro sections for wood anatomical analyses.** Using a circular saw, one radial section capturing all tree rings from the pith to the bark was cut from each stem disks taken at 1.3 m in height. From there these 1-cm wide sections, transverse micro sections of 10–20 μm thickness were cut with a GSL-1 microtome[57]. We applied a cornstarch solution to the sample surface to reduce detached cell walls due to pressure induced by the microtome blade[58]. We then washed the sections with distilled water[59], briefly steamed them to gelatinize any remaining cornstarch granules[60], rinsed them with distilled water to remove the gelatinized cornstarch, and bleached the sections[59]. We followed a standard double-staining procedure[59]: We applied a 1:1 mixture of Safranin and Astrablue to the micro sections for 5 min; washed them with distilled water; and then washed them with ethanol (70, 96 and 99%) to further remove surplus stain. Canada balsam was warmed to temporarily increase viscosity and applied to the micro sections, which were then covered by a glass cover slip. We then heated the micro sections in an oven at 60 °C for a minimum of 8 h to harden the Canada balsam. For illustration purposes, digital photos were taken with a Nikon Eclipse Ni-E upright microscope with a resolution of 5 megapixels and 12-bit color depth (Nikon DS-Fi2). Micrographs were taken at 20× magnification and automatically stitched together with the NIS-Elements software (Version 4.20.01, Nikon Corporation, 2014).

**Measurements of cold damage.** Blue and frost ring occurrences were recorded at the tree-ring level using a light microscope at 100× magnification; sample labels were coded to prevent observer bias. We assigned an intensity score for blue and frost rings on a scale of zero to five (Supplementary Fig 2): A zero indicates absence and a one represents a faint blue band in at least one row of cells, while a five represents a larger band of completely un-lignified cells often associated with narrower and distorted cell walls (Supplementary Fig 2). Frost rings are characterized by deformed tracheids of varying sizes, thin cell walls, and damaged ray parenchyma[19,43]. Frost rings that occurred directly at the ring boundary in the earlywood were labeled as position 1 (Fig. 1a). Their position indicates cold events in winter prior to—or at the time of—cambial reactivation. Frost rings labeled as position 2 occurred after several rows of earlywood tracheids had already formed normally, thereby pointing to a spring cold event after growth had already been initiated (Fig. 1b).

**Climate data**. We derived daily temperature minimums and averages from the ECMWF's ERA interim data for 1979 to 2006[61] for the grid-points closest to our site coordinates. This data is available through KNMI's Climate Explorer (climexp. knmi.nl/). To capture general temperature conditions, as well as cold and frost events on an annual level, we calculated several additional climate variables. The start of the growing season was set as the last day in spring on which minimum temperatures dropped below 0 °C (Julian date). The end of the growing season was set to the Julian day on which the first frost (<0 °C) occurred again in autumn. Growing season length was calculated as the number of Julian days between the start and the end of the growing season. Growing degree days were calculated as degree days above 5 °C to characterize conditions within the growing season. To capture spells of warm weather early in the spring, before the last frosts have occurred, the Julian day of the year at which the sum of growing degree days over 5 °C exceeded 100 was calculated. To investigate the relationship between blue and frost rings and the climate at the seed source locations, we interpolated climate normal data for the 1961–1990 period with the software ClimateNA[62]. This period was chosen because of wide spatial representation of climate stations and because it precedes the most recent anthropogenic climate warming signal.

**Statistical analyses**. The statistical analyses and graphical illustrations were conducted in the R programming environment[63]. Due to the ordinal scale of the blue and frost ring scores, we used a cumulative link mixed model, implemented in the "clmm" function of the "ordinal" package for R[64]. This model was used to analyze differences between populations and to assess relationships with climate. A logit link and flexible thresholds between the ordinal scores was used. Site, block, provenance and tree identifier were specified as random effects modifying the intercepts. The unique tree identifiers were included to account for repeated measures taken on each tree[65]. Population was specified as fixed effect. In addition, several different climate variables were included as fixed effects. Continuous variables were scaled to ensure model convergence and tested for collinearity with Pearson's correlation coefficient. Climate variables were selected based on hypotheses about the cause of blue and frost rings. For blue rings, the Julian dates of the start and end of the growing season as well as the annual sum of growing degree days over 5 °C were tested. For frost rings at position 1, the Julian date of the end of the growing season and the annual sum of growing degree days over 5 °C of the previous year were included. Finally, for frost rings at position 2, the Julian date on which the sum of degree days over 5 °C exceeded 100 and the Julian date of the start of the growing season was tested. Categorical variables were assessed for collinearity with boxplots[65]. Because stem diameter was found to be collinear with population, we tested the effect of diameter in separate models (Supplementary Tables 3–8). Akaike's information criterion was used to select models with good fit and to simplify models. Effects were tested for significance using likelihood ratio tests.

Pairwise testing of the four levels of population for differences was conducted with the "lsmeans"[66] and "multcomp"[67] packages, with an adjustment of p-values following the Tukey method. For illustration, and to account for the slightly imbalanced design and for variation in the length of the tree-ring series as trees reached the sampling height at different ages, least square means of blue and frost ring intensities were calculated with the "lsmeans" function of the "lme4" package[68] after building a linear mixed effects model with the "lmer" function. Similarly, to assess the relationship between blue and frost rings and the normal climate at the provenances' source locations, we calculated least square means of blue and frost ring intensities for the 20 provenances. Subsequently, we calculated Pearson's correlation between climate normals and least square means of blue and frost ring intensities. Rank changes of populations were tested according to the Gail & Simon likelihood ratio test[69] as implemented with the "qualint" function from the "QualInt" package[70]. The Gail Simon likelihood ratio test is a method for testing the significance of qualitative or crossover interactions. A qualitative interaction exists where levels of a group perform better under some circumstances or treatment levels and vice versa. The null hypothesis is that the treatment effects in all sub-groups are in the same direction. In our case, we apply the Gail Simon test to evaluate the significance of rank changes in tree population growth under different time periods (pre- and post-frost in 1992).

**Data availability**. All relevant data are available from the authors.

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

## Acknowledgements

We are grateful to several field and laboratory assistants: J. Braun-Wimmer, A. Bueno, S. Giese, J. Grossmann, M. Harrhy, E. Koertels, J. Rabenschlag, A. Vorländer, and A. Wiegelmann. Thanks also to N. Ukrainetz and V. Berger from the B.C. Ministry of Forests, Lands, Natural Resource Operations and Rural Development (FLNRORD) for site information. We are grateful to FLNRORD for establishing and maintaining valuable field trials and for allowing site access. Funding was provided by NSERC (STPGP-430183, STPGP-494071, CGS-D), DFG (SP 437/18-1), Alexander von Humboldt Foundation (Feodor Lynen Research Fellowship), and Alberta Innovates Technology Futures (Graduate Scholarship).

## Author contributions

D.M. and M.I.R. conceived the study, organized and implemented field work, and collected data. D.M. was responsible for laboratory work. D.M. and M.I.R. conducted analyses and writing, with contributions from A.H. and H.S.

## Additional information

**Competing interests:** The authors declare no competing interests.

