## [Peer Review File · Nature Communications]

Reviewers' comments:

Reviewer #1 (Remarks to the Author):

The manuscript provides an assessment of the occurrence of cold-induced tree-ring anatomical features (namely frost and blue rings) in 20 different provenances of *Pinus contorta* from four different North-West American climatic regions and that have been established within three different common garden experiments. The study aims at inferring information about potential limitations of the use of an assisted migration to mitigate harmful effect of climate change. The three common gardens are located close to each other and are situated well within the distribution area of the species, so that many selected provenances are either facing a northward or a southward shift.

In particular, the authors make use of intra-seasonal datable archive of cold-induced markers in stem tree-rings as an indicator of the adaptability of the provenance to cold events. Two types of tree-ring damaging events are investigated: namely the early and late frost events. A relative higher occurrence of cold-induced damages is considered as an indicator of provenance maladaptation.

Results indicate that Northern provenances had a higher susceptibility to early frosts (higher occurrence of blue first position frost rings) while southern provenances tended to be more exposed to late frost events (Fig 2 1-c). A decline in growth (ring width) of the most southern provenances aftermath a frost event in 1992 has been interpreted as an indicator of maladaptation-induced risk of productivity loss. Correlations between occurrences of cold-induced markers with climatic variables indicated a strong climatic influence on early-frost (blue ring and position 1 frost ring) and less strong with late frosts. Similarly, strong correlations have been found between occurrence of frost damages and with the climate at seed-source locations.

Based on such evidences the authors claim that 1) the approach used of combining retrospective tree-ring analysis on provenance trials is valuable and might deserve to be further applied; and that 2) the impact of an assisted migration to mitigate the effect of climate change based on seed transfer over large environmental gradients might be limited due to increased risk of frost damage and reduced productivity.

The approach applied of using well-documented common garden experiments is a smart way to value years of work and investment of research funding. With this study the authors nicely demonstrate that common garden experiments are an important resource for studying climate change impact on trees and forest ecosystems. The analyses are clear and reproducible, except for the categorization of the intensity of frost damages. The obtained results nicely raise the warning that an assisted migration can also imply harmful responses and thus I am convinced that such a study is an important milestone for shaping a more careful thinking in the field.

However I have some few concerns related to the interpretation of the soundly presented results. Specifically:

1) Figures 2a-c compare the frost-damage occurrence among climatic regions and provenances. The occurrence of the damages of Southern Interior provenances, i.e. the provenances that are very close to the location of the common garden experiment, are among the highest for blue rings and for position 1 frost ring. Since the risk of damage for provenances from more remote climatic regions are inferior of the local adapted ones it suggests that, at least for this type of event, the assisted migration has a beneficial effect. However this statement seems to contradict Authors claim that Southern provenances are at high risk of maladaptation to late frost.

2) A frost damage recorded in the tree-ring anatomy is usually resulting from cold events occurring during the process of tree ring formation (see text at lines 64-65) and can include an single climatic event that occurs on a short time scale (e.g. few hours to few days). What is the rationale of correlating the occurrence of frost damages to multi-monthly average climatic conditions or cumulative degree days (as performed in Table 1)? What are the implications for the interpretation of the results?

3) At line 85 it is stated that "The occurrence of the most severe blue rings in fall was usually followed by spring frost rings at position 1". How are the position 1 frost rings analyzed? Since the origin of

"position 1 frost rings" and "blue rings" are caused by the same type of event (i.e.; early frosts), wouldn't be more appropriate to merge these occurrences in a unique category before performing the analysis?

4) As stated by the authors, the probability to have a cold marker is higher when the tree is young/small (line 91-94). Does this statement affect the interpretation of results showed in Table 1?

Detailed comment:

Ln 20-21: This study does not present any results related to quantitative genetic, while the use of tree-ring analysis in common garden experiment is not novel.

Ln 22: The term "maladaptation" implies a reduced fitness and selective pressure. The results presented in this study can only speculate of the impact of frost damages on the provenances, since there are no sign of higher mortality and/or the occurrence of damages is inferior of than in local provenances.

Ln 24: the term "rank-changing growth loss" sounds weird

Ln 63-64. Is it correct to state that frost rings are related to generally colder year? It is however expectable that in colder year the risk of frost event is higher, and thus I would propose to only keep the second part of the sentence.

Ln 73: Here it would be appropriate to provide more information regarding the study design (mentioning the 20 population grouped in 4 homogenous source regions planted in three common gardens in the center of the current distribution area of the species).

Ln 73: I propose to change "cold adaptation" to "susceptibility to frost events".

Ln 73: The term "population" assumes different meaning throughout the manuscript and it creates confusion. Here it refers to common gardens, at lines 118/120 it refers to provenances.

Ln 75: the climate pattern are referred to climate at the source location or at the location of the common gardens?

Ln 77 the expression "in arguably" sounds weird in this context

Ln 82 "In total, we examined 2999 tree rings from 117 trees from 20 different provenances grouped in 4 regions."

Ln 83: Across all populations, we recorded the occurrence of 689 blue ring (23%) and 204 frost ring (7%).

"Ln 92-94: "This may be due to their smaller size and lower heat absorption, higher occurrences of tender lammas growth, ...". Add "occurrences". In addition what is "lammas"?"

Ln 95: "An analysis of historical climate data at the common garden sites reveals ..."

Ln 97: remove the reference to the figures (Fig 1d, 1e) since they do not provide clear evidence of the mentioned link

Ln 98-99": I am not sure that this statement is correct "Cold spring conditions delay the onset of growth, pushing growth later into the fall. " I do not think that there is a direct link between the onset and the length of the growing season.

Ln 110: The figure 1f shows a short warm spell is in summer (mainly around July). Does this timing fit the timing when the earlywood is in formation?

Ln 113: Tab 1 does not show differences among genotypes (provenances) but only among regions

Ln 116: Which regions/provenances resulted to be significantly different?

Ln 121-123: "In contrast, Northern populations were the most susceptible group to spring frost damage as indicated by frost rings in position 2". It seems to be mostly due to only the provenance P33.

Ln 137: The citation 48 refers to frost crack that is a different damage from frost ring or blue ring:

Ln 194-197 and 199-200: Not clear on what material (cores, micro-section or images of micro-sections) you have performed the observation. If on micro-slides, why then taking images? What did the images have been used for?

Ln 200-203: there are no results related to intensity scores. Why then creating this categories. In addition, a occurrence of 23% of blue rings sound to be very high, I therefore would be interested to

see an example of lower category of blue ring (maybe this can be add in the Supplementary material) 215-219: the definition of growing season is based on purely climatic criteria. In which sense is this definition appropriate for tree growth?

Figure 1: change "grey bars indicate" to "grey bar indicates". This mistake occurs several times in the figure legend.

Figure 2d-f: What does the size of the dots represent?

Table 1: The amount of frost damages in the manuscript is either termed "intensity" or "frequency", while it would be more appropriate to use the term "occurrence". Consider adding what is the number of samples n in each correlation.

Table 2: Consider adding the R2 ... and the number of samples n in each correlation

Figure S2 (In 508-509): not clear what is meant with "Spearman correlation coefficients for height and diameter and each type of cold imprint are in the lower right corner of each panel. "

Sincerely
Patrick Fonti

Reviewer #2 (Remarks to the Author):

I am enthusiastic about the major structural features of this paper. The question asked (and answered) is interesting and important – i.e., what are some of the potential pitfalls of an assisted migration strategy towards forest "adaptation" (not in the evolutionary biology sense, in the forest management sense) to climate change. The Introduction very nicely and convincingly lays out the issues. Further, the combination of tree-ring research and provenance trials is a powerful and novel one. In these two respects, to the best of my knowledge, the paper represents something both novel and of importance. The improved understanding of the causes of the three types of cold damage that the authors lay out is, to the best of my knowledge, also a new contribution (which will be appreciated by a smaller, more technical audience, but is very valuable contribution...in the service of a broad and important question).

My primary criticisms of the paper, which I hope will be constructive, concern the statistical analyses. One of the two statistical analyses performed is deeply flawed because it does not reflect the structure of the data. Note that there are two responses analyzed, both derived from the same data set (i.e., same experimental design): 1) the occurrence of damaged rings (blue rings and two types of frost rings) and 2) among those rings with evidence of damage, the intensity of damage (intensity being a categorical metric ranging from one to five). The authors have (rightly so) used a generalized linear mixed model (glmm) to analyze the former, with region treated as a fixed effect and several other factors treated as random effects (provenance, site, tree, and year). The glmm accounts for clustering in the data that effectively reduces the sample size. The same issue of clustering/pseudoreplication needs to be dealt with in the analysis of the "intensity" variable. Further, it is not clear if the authors have averaged across trees to convert the response, which is categorical, to a response that is continuous to be able to use a correlation analysis, and how that averaging was done, or if they've simply treated the intensity categories as if they were continuous and run a correlation on the raw values? It would be more appropriate/statistically powerful to use a multinomial model to analyze a categorical response – the nature of the response is multinomial.

A second point about the statistical analyses is that many, many climate variables are tested as predictors. I'm not so concerned about the multiple tests issue; rather, I would like to see more thought put into the nature of clustering of variation in climate across those multiple dimensions. Climate is by nature a multivariate beast. A principle components analysis of the climate data would allow the authors (and readers) to better understand collinearity between the many predictors tested. Is a year with an early start to the growing season warmer than average throughout the growing season? What <combination> of climatic conditions is most strongly associated with the three types of

cold damage within the observation period? If there is strong clustering among the dimensions (i.e., PC1 + 2 + 3 explains a lot of the temporal variation in climate), then it makes sense to either choose one predictor from each cluster, or use the PC's themselves as predictors. See Dormann et al. 2012 *Ecography* for these and other strategies regarding analyses in the presence of collinearity.

A third issue is that Figure 1d makes it clear (and the authors conclude/discuss) that the incidence of frost rings in position 1 is influenced by the age/size of trees. It is a form of cold damage that is evidently much more prevalent when trees are small. This should be modeled explicitly in the glmm of frost ring position 1 occurrence (and tested regarding the other two damage types). Finally, there is no reason that climate predictors could not be included explicitly in the glmm's (logistic regressions) of damage occurrence.

Additional comments that I hope will improve the presentation/impact of the research follow:

In contrast to my comments above about the strong framing of the paper (i.e., in terms of assisted migration as a strategy to address forest maladaptation, tree rings as a record of risks associated with assisted migration), the research question as it is stated at the end of the Introduction is weak "Here,...we first test whether frost imprints in tree rings can act as a reliable record of cold maladaptation..." (why are there no line numbers associated with this manuscript?). This is a very limited question compared to the rest of the Introduction. I also found the phrase "...in arguably the largest provenance trial series in existence" to be unnecessarily overstated, since the data presented in this manuscript were a limited subset of the Illingworth provenance trial. There is no need to bluff a larger design than was actually analyzed.

Another point is the second-to-last sentence in the abstract concerns a pattern that does not deserve to be highlighted in the abstract. The rank-order change of growth of the US populations following the frost event in spring, 1992 is <not> analyzed statistically (there are no error bars associated with the growth trends in Figure 2d-f), and it reverses by the end of the time series (2005), suggesting that after a long period of relatively poor growth, those trees may have been bouncing back. Another statistical analysis would need to be done to properly analyze these trends (see Lloret et al. 2011 *Oikos* paper, and the R package *pointRes* [van der Maaten-Theunissen et al. 2015 *Dendrochronologia*] that implements the resistance/resilience/recovery metrics defined by Lloret et al).

The most interesting result of the paper should be highlighted more strongly as a take-away message: there are opposing geographical clines of susceptibility to frost damage incurred by early-season frosts ("false spring", frost ring position 2) vs. end-of-season frosts (blue rings and frost rings position 1), with northern provenances more susceptible to the former and southern provenances more susceptible to the latter. (By the way, the interpretation of this result is convincing and nicely argued.) Additionally, there is a parallel geographical cline with respect to absolute growth rate. These opposing geographic clines (particularly re: two kinds of frost damage) should be more strongly highlighted in the abstract (in the place of the sentence about rank-order change of growth of the southern-most populations) as a serious challenge to assisted migration strategies. I can only imagine that the negative evolutionary correlation uncovered by the authors in this study is the tip of the iceberg in terms of how assisted migration may fail (many other such negative evolutionary correlations are probably out there waiting to be discovered that constrain the success of assisted migration).

I have additional detailed comments, but without line numbers it is difficult to provide those.

This is a nice study overall with interesting and important results, but needs to be re-analyzed to properly reflect the study design.

Reviewer #3 (Remarks to the Author):

Montwe et al. analyse the effects of frost and extreme cold events at the beginning and the end of the vegetation period on growth and wood formation in trees. The authors make use of old provenance experiments which enable them to test for differences between 20 populations originating from major parts of the species (lodgepole pine) natural range and which allow for understanding effects on frost

events under in situ conditions. They demonstrate that staining wood discs with a double staining procedure allowed identifying deformed tree rings and rings with incomplete lignified cells, and that these wood deformations can be clearly linked to frost events either at the beginning or the end of the vegetation period. The frequency of deformed rings differed significantly among populations and population groups and allowed to estimate effects of seed transfer on wood development and annual growth.

Overall, the paper is well written, methodological sound and provides a fascinating new procedure to make use of tree provenance trials for climate change adaptation research. The methodology provided will certainly be adapted by forest scientist around the world for similar questions. Just by reading the paper, I found that the methodology might also support scientist to understand the risks associated with extended vegetation periods in climate change.

The general topic of the paper is highly relevant: after about a decade of research and discussion of pro and cons of assisted migration in forest trees, assisted migration is now being integrated into reforestation and restoration practice. The given manuscript is highly needed as it sheds light on the potential drawback of artificial tree movement if populations are being moved too far from the climate conditions to which they are adapted to. This kind of risk assessment is required for seed transfer models. Thus, I see the paper as relevant and expect it to have a high impact on forest regeneration policies in northern boreal and temperate forests, where frost events will also in climate change be a significant driver of vegetation development. Given the high public and political interest in future forest development, the paper should be considered as highlight paper in the New and Views of the Nature. Overall, I highly recommend the publication of the given manuscript.

Anyhow, there are a few minor issues that the authors should consider:

L131: "... northern populations are adapted to ..." instead of "must adapt"

L141ff: I was surprised that the southern interior populations showed the highest fall frost damages, as these populations originate from the region where the three study sites are being located, so they should be locally well adapted. It was just when I checked tables S1 and S2 that I found that the test sites are located 300-800 m higher than the seed origins in southern interior. This certainly explains the higher late frost damages. Please, add few lines about the higher location of test sites to the discussion.

L223: what was interpolated? What interpolation technique was used?

Figures S1: its difficult to identify what the r in the Figure mean. I rather suggest a table clearly indicating which traits are being regressed to each other and giving the full stats of the regression (r, p-value, etc.)

L236: here the full mixed model is being described. However, I cant see the full result. Please add another suppl. table with the complete stats for each of the tested effects.

Please find our replies to specific reviewer comments inserted below in blue.

Reviewer #1 Comments:

Comment 1.1: The manuscript provides an assessment of the occurrence of cold-induced tree-ring anatomical features (namely frost and blue rings) in 20 different provenances of *Pinus contorta* from four different North-West American climatic regions and that have been established within three different common garden experiments. The study aims at inferring information about potential limitations of the use of an assisted migration to mitigate harmful effect of climate change. The three common gardens are located close to each other and are situated well within the distribution area of the species, so that many selected provenances are either facing a northward or a southward shift.

In particular, the authors make use of intra-seasonal datable archive of cold-induced markers in stem tree-rings as an indicator of the adaptability of the provenance to cold events. Two types of tree-ring damaging events are investigated: namely the early and late frost events. A relative higher occurrence of cold-induced damages is considered as an indicator of provenance maladaptation.

Results indicate that Northern provenances had a higher susceptibility to early frosts (higher occurrence of blue first position frost rings) while southern provenances tended to be more exposed to late frost events (Fig 2 1-c). A decline in growth (ring width) of the most southern provenances aftermath a frost event in 1992 has been interpreted as an indicator of maladaptation-induced risk of productivity loss. Correlations between occurrences of cold-induced markers with climatic variables indicated a strong climatic influence on early-frost (blue ring and position 1 frost ring) and less strong with late frosts. Similarly, strong correlations have been found between occurrence of frost damages and with the climate at seed-source locations.

Based on such evidences the authors claim that 1) the approach used of combining retrospective tree-ring analysis on provenance trials is valuable and might deserve to be further applied; and that 2) the impact of an assisted migration to mitigate the effect of climate change based on seed transfer over large environmental gradients might be limited due to increased risk of frost damage and reduced productivity.

The approach applied of using well-documented common garden experiments is a smart way to value years of work and investment of research funding. With this study the authors nicely demonstrate that common garden experiments are an important resource for studying climate change impact on trees and forest ecosystems. The analyses are clear and reproducible, except for the categorization of the intensity of frost damages. The obtained results nicely raise the warning that an assisted migration can also imply harmful responses and thus I am convinced that such a study is an important milestone for shaping a more careful thinking in the field. However I have some few concerns related to the interpretation of the soundly presented results.

Thank you for your assessment. Please see below for descriptions of how we have addressed your concerns.

Comment 1.2: 1) Figures 2a-c compare the frost-damage occurrence among climatic regions and provenances. The occurrence of the damages of Southern Interior provenances, i.e. the provenances that are very close to the location of the common garden experiment, are among the highest for blue rings and for position 1 frost ring. Since the risk of damage for provenances from more remote climatic regions are inferior of the local adapted ones it suggests that, at least for this type of event, the assisted migration has a beneficial effect. However this statement seems to contradict Authors claim that Southern provenances are at high risk of maladaptation to late frost.

Thank you for this comment. The geographically close provenances were sourced from lower elevation and therefore higher in mean annual temperature than the planting sites. Thus, the provenances from the southern interior represent a climate transfer, explaining the higher occurrence of blue and frost rings in these provenances.

To clarify, we added a new panel to Figure 2, which was initially left empty for the caption but was space that could be filled with a new plot. We now show the transfer not only in terms geography (i.e. Fig 2g) but also in terms of mean annual temperature between the provenances and the average site conditions (vertical line in Fig 2h).

We further revised the relevant manuscript section to read as follows:

“Another equally important finding is that it may not always be possible to transfer genotypes northward or upward in elevation as a management prescription to address climate change: Both the US populations and the Southern Interior populations were transferred to colder environments in this experiment (Fig 2h), with the test sites located either further north or higher in elevation. The productive Southern Interior populations showed the highest fall frost damage (Fig 2d-e, SI a-b). Given that short growing seasons and low cumulative degree days are a likely cause of blue rings and frost rings in fall, a transfer of these seed sources to colder environments is likely to exacerbate this problem.” (Lines 139-146)

Comment 1.3: 2) A frost damage recorded in the tree-ring anatomy is usually resulting from cold events occurring during the process of tree ring formation (see text at lines 64-65) and can include an single climatic event that occurs on a short time scale (e.g. few hours to few days). What is the rational of correlating the occurrence of frost damages to multi-monthly average climatic conditions or cumulative degree days (as performed in Table 1)? What are the implications for the interpretation of the results?

It is true that frost damage can occur after a specific frost event. We found, however, that the main driver of blue and frost rings is a misalignment of growth and the start and end of the growing season. When analyzing the daily temperature data, we found multiple frost events that could potentially cause blue and frost rings. However, these events appear only consequential if 1) trees started growing early in the spring and 2) trees kept growing late in the fall. Therefore,

the analysis with growing degree days and seasonal climate descriptors adds value here. We more clearly point it out in this revised paragraph in the result section:

“An analysis of historical climate data at the common garden sites reveals that blue rings and frost rings in position 1 are linked to a late initiation of the growing season, low temperatures at the end of the growing season, and a generally cool growing season (Tab 1, Fig 1d, 1e). Because lignification can lag behind cellular development by several weeks^{32,33,42–44}, the growing season length and overall growing conditions during the season have an impact on the occurrence of frost and blue rings if the lignifications process is incomplete by the time the first fall frosts occur.” (Lines 91-97)

Comment 1.4: 3) At line 85 it is stated that “The occurrence of the most severe blue rings in fall was usually followed by spring frost rings at position 1“. How are the position 1 frost rings analyzed? Since the origin of “position 1 frost rings” and “blue rings” are caused by the same type of event (i.e.; early frosts), wouldn’t be more appropriate to merge these occurrences in a unique category before performing the analysis?

This is a valid point, however, because there are two features showing different patterns in the xylem, and because they appear in different parts of the tree-rings (early and latewood), we decided to present them separately but discuss the relationship in the manuscript.

Please also refer to the clarification above (comment 1.3). Blue rings may simply represent a premature ending of the lignification process that could not be completed in a short and/or cool growing season (i.e. we have blue rings without frost rings in position 1). That said, frost rings in position 1, representing a fall frost event, are more likely if the tree is still biologically active at the end of the growing season, completing lignification processes at the end of a very short/cold growing season. By analyzing them separately, we could show that the relationships are similar, thereby pointing to the link itself.

Comment 1.5: 4) As stated by the authors, the probability to have a cold marker is higher when the tree is young/small (line 91-94). Does this statement affect the interpretation of results showed in Table 1?

This is a good point. The trend towards higher susceptibility to frost damage in early years could be confounded with age, and this is accounted for with a random effect for Year (=Age) in statistical testing. In addition, years with contrasting or high frequencies also occur later in the series (e.g. 1996, 1998/1999, 2003), which are related to the length and warmth of the growing season. This indicates that these relationships remain true despite potential higher vulnerability at a younger age. It is also worth noting that the early years with higher frequencies (fig. 1d) are also cool years with shorter growing seasons (fig. 1e).

Comment 1.6: Ln 20-21: This study does not present any results related to quantitative genetic, while the use of tree-ring analysis in common garden experiment is not novel.

Studying frost ring signatures in provenance trials is certainly novel, but we address this concern early in the abstract, where we also draw attention on the key finding of opposing clines. The revised sentence in the abstract reads:

“Using tree-ring signatures of cold damage in common garden trials designed to study genetic population differentiation, we find opposing geographic clines for spring frost and fall frost damage.” (Lines 5-7)

Comment 1.7: Ln 22: The term “maladaptation” implies a reduced fitness and selective pressure. The results presented in this study can only speculate of the impact of frost damages on the provenances, since there are no sign of higher mortality and/or the occurrence of damages is inferior of than in local provenances.

Thank you. Our intended meaning for maladaptation is that historic climate adaptations are now becoming inadequate, which is a commonly adopted definition in genecology. However, it is a good point that we do not have data to make a statement regarding fitness. We have therefore rephrased this sentence to:

“Provenances from the northern extent of the species range are sensitive to false springs.” (Lines 7-8)

Comment 1.8: Ln 24: the term “rank-changing growth loss” sounds weird

Thank you, we have modified this sentence as follows:

“Transferring the southern, warm-adapted genotypes northward caused a significant loss of growth and permanent rank change after a spring frost event.” (Line 9-11)

In response to the second reviewer, we have also added a statistical test to back this statement up (see comment 2.7).

Comment 1.9: Ln 63-64. Is it correct to state that frost rings are related to generally colder year? It is however expectable that in colder year the risk of frost event is higher, and thus I would propose to only keep the second part of the sentence.

Thanks. While the cold event itself is a driver, this is also linked to generally cooler shoulder seasons, overall length and growing season warmth. To clarify, we rephrase to:

“Frost rings have been linked to generally colder years³² in which air temperatures fall below freezing during a period of cambial activity¹⁶”(Lines 49 – 50)

Comment 1.10: Ln 73: Here it would be appropriate to provide more information regarding the study design (mentioning the 20 population grouped in 4 homogenous source regions planted in three common gardens in the center of the current distribution area of the species).

The rephrased sentences now read:

“A subset of 20 provenances across western Canada and the United States, grown at three central planting sites, were selected to represent a wide range of climate transfers.” (Lines 69-71)

and

“In total, we examined 2999 tree rings from 117 trees, representing 20 provenances grouped into four geographic regions” (Lines 76-77)

Comment 1.11: Ln 73: I propose to change “cold adaptation” to “susceptibility to frost events”.

Rephrased:

“Here, we study these cold and frost signatures in trees grown in provenance trials, allowing us to quantify genetic differentiation in susceptibility to cold among populations.” (Line 59)

Comment 1.12: Ln 73: The term “population” assumes different meaning throughout the manuscript and it creates confusion. Here it refers to common gardens, at lines 118/120 it refers to provenances.

Thank you for pointing out this possible point of confusion for readers. To clarify this throughout the manuscript, we are now internally consistent with the term “population” referring only to the regional populations. Any time we are discussing seed collection sites, we now only use the terms “provenances”, “genotypes” or “seed sources” (these are seed sources within regional populations).

Comment 1.13: Ln 75: the climate pattern are referred to climate at the source location or at the location of the common gardens?

Thank you. Fixed as follows:

“We first investigate whether frost imprints in tree rings can act as a reliable indicator of cold susceptibility by analyzing the association of climate and cold events at the test sites with the occurrence of blue and frost rings.” (Lines 65-67)

Comment 1.14: Ln 77 the expression “in arguably” sounds weird in this context

We rephrase this sentence to:

*“Our study species is lodgepole pine (*Pinus contorta* Dougl. ex. Loud.), with a range-wide collection of seed sources grown for three decades in the Illingworth provenance trial³⁷.”* (Lines 61-63)

Comment 1.15: Ln 82 “In total, we examined 2999 tree rings from 117 trees from 20 different provenances grouped in 4 regions.”

Changed as suggested. This also addresses comment 1.10. Please see quote there.

Comment 1.16: Ln 83: Across all populations, we recorded the occurrence of 689 blue ring (23%) and 204 frost ring (7%).

Thank you for the suggested rephrasing, which we have now adopted (Lines 77-78).

Comment 1.17: “Ln 92-94: “This may be due to their smaller size and lower heat absorption, higher occurrences of tender lammas growth, ...”. Add “occurrences”. In addition what is “lammas”?”

The sentence was simplified as follows:

“Higher occurrences of blue and frost rings in the earlier part of the experiment (Figs 1d) may reflect higher vulnerability of small saplings to frost (Supplementary Fig 2). This may be due to their smaller size and lower heat absorption, more tender growth from second flushing in younger trees, and their proximity to the soil surface, which is colder due to radiative cooling^{31,38-41}.” (Lines 85-89)

Comment 1.18: Ln 95: “An analysis of historical climate data at the common garden sites reveals ...”

Thank you. Re-worded to include “at the common garden sites”, as suggested (Line 91).

Comment 1.19: Ln 97: remove the reference to the figures (Fig 1d, 1e) since they do not provide clear evidence of the mentioned link

The length of the bars in Fig 1e indicates the length of the growing season and the shading of the bar refers to the accumulation of warmth in that growing season. Since the x-axis in both panels 1d (occurrence of blue and frost rings) and panel 1e (growing season length/warmth) is the same, one can compare the blue and frost ring occurrences to the climate conditions in that given year.

To clarify that the table shows the analysis and the figures are for a visual check, we now rephrase to:

“An analysis of climate data at the common garden sites reveals that blue rings and frost rings in position 1 are linked to a late initiation of the growing season, low temperatures at the end of the growing season, and a generally cool growing season (Tab 1, also apparent by comparing Figs 1d, 1e).” (Lines 91-94)

Comment 1.20: Ln 98-99”: I am not sure that this statement is correct “Cold spring conditions delay the onset of growth, pushing growth later into the fall. “ I do not think that there is a direct link between the onset and the length of the growing season.

We rephrase to clarify (also in accordance with comment 1.3):

“Because lignification can lag behind cellular development by several weeks^{32,33,42-44}, the growing season length and overall growing conditions during the season have an impact on the occurrence of frost and blue rings if the lignifications process is incomplete by the time the first fall frosts occur.” (Lines 94-97)

Comment 1.21: Ln 110: The figure 1f shows a short warm spell is in summer (mainly around July). Does this timing fit the timing when the earlywood is in formation?

Thank you for pointing out this ambiguity. In this figure we refer to the warm spell in April. We rephrase to clarify:

“As an example, the year 1992 had a high proportion of frost rings in position 2, and a false spring: a warm spell in April around day 100 (Fig 1f).” (Lines 105-107)

Comment 1.22: Ln 113: Tab 1 does not show differences among genotypes (provenances) but only among regions

This sentence was removed.

Comment 1.23: Ln 116: Which regions/provenances resulted to be significantly different?

Thank you. We added a table in the Supplementary Table 3 that shows (adjusted) multiple comparisons of population’s occurrences of blue and frost rings. We also added a reference to this table in Line 114-115 before discussing regional differences.

Comment 1.24: Ln 121-123: “In contrast, Northern populations were the most susceptible group to spring frost damage as indicated by frost rings in position 2”. It seems to be mostly due to only the provenance P33.

Yes, notably the most northern source from the coldest location:

“Susceptibility of populations to cold damage under increased spring temperature variability appears to be most pronounced in the North, with the most northern seed source and coldest location of origin showing the highest occurrence of frost rings in position 2 (Fig 2g and 2h)” (Lines 108-111)

Comment 1.25: Ln 137: The citation 48 refers to frost crack that is a different damage from frost ring or blue ring:

Thank you. We rephrase to:

“Moreover, the resulting frost damage may be associated with forest health issues. Such damage could also reduce wood quality and value by creating weaknesses and defects in the timber⁴⁸.” (Line 136-137)

Comment 1.26: Ln 194-197 and 199-200: Not clear on what material (cores, micro-section or images of micro-sections) you have performed the observation. If on micro-slides, why then taking images? What did the images have been used for?

Descriptions of the materials and the purpose of the images have been added:

“Using a circular saw, one radial section capturing all tree rings from the pith to the bark was cut from each stem disks taken at 1.3 m in height. From there these 1 cm wide sections, transverse micro sections of 10-20 μ m thickness were cut with a GSL-1 microtome⁵⁰” (Lines 185-187)

“For illustration purposes, digital photos were taken with a Nikon Eclipse Ni-E upright microscope with a resolution of 5 megapixels and 12-bit color depth (Nikon DS-Fi2). Micrographs were taken at 20 \times magnification and automatically stitched together with the NIS-Elements software (Version 4.20.01, Nikon Corporation, 2014).” (Lines 196-200)

Comment 1.27: Ln 200-203: there are no results related to intensity scores. Why then creating this categories. In addition, a occurrence of 23% of blue rings sound to be very high, I therefore would be interested to see an example of lower category of blue ring (maybe this can be add in the Supplementary material)

Thank you. Genetic differences among cold damage were indicated using the intensity score in Figures 2 a-c, and the intensity scores are used for correlations (Tabs 1, 2). The high number of blue rings may be explained by the high number of southern provenances transferred to the colder experimental sites.

As suggested, however, we have now added examples of all categories in a figure in the supplementary information section (Supplementary Fig 2).

Comment 1.28: 215-219: the definition of growing season is based on purely climatic criteria. In which sense is this definition appropriate for tree growth?

This definition is indeed exclusively climatic and could also be called frost free period, as also found widely in the literature. While it is true that species and populations differ in their spring and fall phenology, this is highly correlated with cumulative degree days or onset of the frost free period.

Comment 1.29: 215-219: Figure 1: change “grey bars indicate” to ”grey bar indicates”. This mistake occurs several times in the figure legend.

Thank you for pointing this out; changed to “Each grey bar indicates sample size per year” in the figure legend.

Comment 1.30: Figure 2d-f: What does the size of the dots represent?

We revised the legend to explain:

“Figure 2. Blue and frost ring intensities by population and associated annual growth. Panel a) to c) show the distribution and medians of blue ring and frost ring intensities, where provenances are sorted by the mean annual temperature of their source climate (warmest on the left, n=117). Panels d) to f) show how growth is related to blue and frost ring intensities on an annual basis, where average blue and frost ring intensities (including zero values) are represented by the size of the circles. Provenances are colored according to their region and labeled by their identification number. Panel g) shows the location of provenances and regions as well as the species range of lodgepole pine (dark gray) and the location of test sites sampled in this study (black triangles). Panel h) shows the difference of the provenance source climate to the average climate of the test sites thereby indicating the degree and direction of climate transfers.”

Comment 1.31: Table 1: The amount of frost damages in the manuscript is either termed “intensity” or “frequency”, while it would be more appropriate to use the term “occurrence”. Consider adding what is the number of samples n in each correlation.

We went through the manuscript and made sure that the terms “intensity” and “occurrence” are used correctly.

Comment 1.32: Table 2: Consider adding the R2 ... and the number of samples n in each correlation

Sincerely
Patrick Fonti

Thank you for the suggestion. We added the number of samples for the correlation to the description:

“... The correlation of average blue ring and frost ring intensities per regional population are based on 78 values (3 sites × 26 years, from 1980 to 2005, inclusive).”

And:

“...The correlation of average blue ring and frost ring intensities are based on 20 values (climate conditions at the origin of 20 seed collection sites).”

Comment 1.33: Figure S2 (ln 508-509): not clear what is meant with “Spearman correlation coefficients for height and diameter and each type of cold imprint are in the lower right corner of each panel. “

In this supplementary figure (now Figure S1), we show how height and diameter of the provenances are related to blue and frost rings. The correlations show that taller and thicker trees are more affected by blue and frost rings at position 2, but less affected by frost rings at position 1. We have modified the caption for clarify:

“Spearman correlation coefficients are found in the lower right corner of each panel. These correlations are between height or diameter and, depending on the panel, one of the three measures of cold (blue ring intensity, frost ring in position 1, frost ring in position 2)”

Reviewer #2 Comments

Comment 2.1: I am enthusiastic about the major structural features of this paper. The question asked (and answered) is interesting and important – i.e., what are some of the potential pitfalls of an assisted migration strategy towards forest “adaptation” (not in the evolutionary biology sense, in the forest management sense) to climate change. The Introduction very nicely and convincingly lays out the issues. Further, the combination of tree-ring research and provenance trials is a powerful and novel one. In these two respects, to the best of my knowledge, the paper represents something both novel and of importance. The improved understanding of the causes of the three types of cold damage that the authors lay out is, to the best of my knowledge, also a new contribution (which will be appreciated by a smaller, more technical audience, but is very valuable contribution...in the service of a broad and important question).

Thank you for the favorable assessment.

Comment 2.2: My primary criticisms of the paper, which I hope will be constructive, concern the statistical analyses. One of the two statistical analyses performed is deeply flawed because it does not reflect the structure of the data. Note that there are two responses analyzed, both derived from the same data set (i.e., same experimental design): 1) the occurrence of damaged rings (blue rings and two types of frost rings) and 2) among those rings with evidence of damage, the

intensity of damage (intensity being a categorical metric ranging from one to five). The authors have (rightly so) used a generalized linear mixed model (glmm) to analyze the former, with region treated as a fixed effect and several other factors treated as random effects (provenance, site, tree, and year). The glmm accounts for clustering in the data that effectively reduces the sample size. The same issue of clustering/pseudoreplication needs to be dealt with in the analysis of the “intensity” variable. Further, it is not clear if the authors have averaged across trees to convert the response, which is categorical, to a response that is continuous to be able to use a correlation analysis, and how that averaging was done, or if they’ve simply treated the intensity categories as if they were continuous and run a correlation on the raw values? It would be more appropriate/statistically powerful to use a multinomial model to analyze a categorical response – the nature of the response is multinomial.

Thank you for the suggestion. We implemented a multinomial model with ASREML-w, but for non-parametric, ordinal data (i.e. with a meaningful order from 1-5), a repeated measures permutational ANOVA turned out slightly more powerful. This method is suitable for this data because it makes no assumptions about the distribution of the response variable. We include a new table in the supplementary section showing multiple comparisons for both occurrence and intensity (also requested below):

Table S3. Multiple comparisons among populations’ blue and frost ring occurrences

Test	Blue ring		Frost Ring			
	Occurrence	Intensity	Position 1	Position 1	Position 2	Position 2
N - CI == 0	< 0.001	< 0.001	0.968	0.191	1.000	0.603
SI - CI == 0	0.012	< 0.001	0.010	< 0.001	0.416	0.458
US - CI == 0	0.169	0.003	0.304	0.005	0.060.	0.281
SI - N == 0	< 0.001	< 0.001	0.038	< 0.001	0.382	0.374
US - N == 0	< 0.001	< 0.001	0.571	< 0.001	0.053.	0.014
US - SI == 0	0.766	0.082.	0.518	0.016	0.698	0.460

Post-hoc tests adjusted with the Benjamini and Hochberg method.

The method section has been updated as well:

“To test for differences in the intensity score, we used a permutational repeated measures ANOVA⁶², implemented in the lperm package for R⁶³. This method was appropriate because it makes no assumptions about the distribution of the response. We tested the effect of region and site, while accounting for the repeated measures in the error term.” (Lines 246-250)

For the correlation analysis, we did indeed treat the intensity scores as continuous variables, as this is a common approach. Because the time series of the intensity values differ in length among trees, we used last square means, calculated from a linear mixed model, to get an unbiased estimate. This is explained in the following paragraph:

“To account for the slightly imbalanced design and for variation in the length of the tree-ring series as trees reached the sampling height at different ages, the least square means of blue and frost ring intensities were calculated with the lsmeans function of the lme4 package after building a linear mixed effects model with the lmer function.” Lines (250-253)

Comment 2.3: A second point about the statistical analyses is that many, many climate variables are tested as predictors. I'm not so concerned about the multiple tests issue; rather, I would like to see more thought put into the nature of clustering of variation in climate across those multiple dimensions. Climate is by nature a multivariate beast. A principle components analysis of the climate data would allow the authors (and readers) to better understand collinearity between the many predictors tested. Is a year with an early start to the growing season warmer than average throughout the growing season? What <combination> of climatic conditions is most strongly associated with the three types of cold damage within the observation period? If there is strong clustering among the dimensions (i.e., PC1 + 2 + 3 explains a lot of the temporal variation in climate), then it makes sense to either choose one predictor from each cluster, or use the PC's themselves as predictors. See Dormann et al. 2012 Ecography for these and other strategies regarding analyses in the presence of collinearity.

It is true that climate is multivariate and highly collinear. That said, multivariate plots are not always easy to interpret and in our experience there is a danger of losing general readership. Nevertheless, we explored this and we include an example here:

The longest vectors in the direction of the big circles (i.e. high blue ring intensity) are First Autumn Frost <-5°C (FEV_5), and Start of Growing Season (SGS). They also show up as is the highest positive correlation with blue ring intensity in Table 1). Many variables describing a warm and long growing season are autocorrelated pointing away from the largest dots (blue negative correlations in the top panel of Table 1). It is interesting to see FEV_5 and SGS being orthogonal, but the essentials that matter most for supporting the main conclusions of the paper are more space-efficiently conveyed in Table 1.

Comment 2.4: A third issue is that Figure 1d makes it clear (and the authors conclude/discuss) that the incidence of frost rings in position 1 is influenced by the age/size of trees. It is a form of cold damage that is evidently much more prevalent when trees are small. This should be modeled

explicitly in the glmm of frost ring position 1 occurrence (and tested regarding the other two damage types).

Yes, this is true. We do include Year as a random effect in the model. Please also note that most of the responses are driven by inter-annual variation, however, not age. In fact the two most contrasting years that we use as examples in Figure 1f and 1g are adjacent to each other: 1998 and 1999.

Comment 2.5: Finally, there is no reason that climate predictors could not be included explicitly in the glmm's (logistic regressions) of damage occurrence.

This is true, but we prefer to use the glmm and permutational anova for testing genetic differentiation among populations. It does not make interpretation easier to include too many terms, and we already have a separate analysis that addresses this aspect.

Comment 2.6: Additional comments that I hope will improve the presentation/impact of the research follow:

In contrast to my comments above about the strong framing of the paper (i.e., in terms of assisted migration as a strategy to address forest maladaptation, tree rings as a record of risks associated with assisted migration), the research question as it is stated at the end of the Introduction is weak "Here,...we first test whether frost imprints in tree rings can act as a reliable record of cold maladaptation..." (why are there no lines numbers associated with this manuscript?). This is a very limited question compared to the rest of the Introduction. I also found the phrase "...in arguably the largest provenance trial series in existence" to be unnecessarily overstated, since the data presented in this manuscript were a limited subset of the Illingworth provenance trial. There is no need to bluff a larger design than was actually analyzed.

We expected the line numbers to be generated automatically through the submission system, but now include them in the word document.

We also dropped the "arguably largest" phrase, and added more elements to this section also in response to the other reviewer's comments. We now include essential method information in the briefest way, so that our sampling design is understood before the reader moves on to the subsequent result section. The revised paragraph reads:

*"Here, we study these cold and frost signatures in trees grown in provenance trials, allowing us to quantify genetic differentiation in susceptibility to cold among populations. This can inform limits to seed transfers designed to address projected climate change. Our study species is lodgepole pine (*Pinus contorta* Dougl. ex. Loud.), represented by a range-wide collection of seed sources grown for three decades in the Illingworth provenance trial series³⁷. A subset of 20 provenances from across western Canada and the United States, grown at three central planting sites, were selected to analyze the response to climate transfers. We first investigate whether frost imprints in tree rings can act as a reliable indicator of cold susceptibility by analyzing the*

association of climate and cold events at the test sites with the occurrence of blue and frost rings. We further analyze frost imprints in the context of long-term growth data to quantify potential impacts of cold damage on productivity. A subset of 20 provenances across western Canada and the United States, grown at three central planting sites, were selected to represent a wide range of climate transfers.” (Lines 59-73)

Comment 2.7: Another point is the second-to-last sentence in the abstract concerns a pattern that does not deserve to be highlighted in the abstract. The rank-order change of growth of the US populations following the frost event in spring, 1992 is <not> analyzed statistically (there are no error bars associated with the growth trends in Figure 2d-f), and it reverses by the end of the time series (2005), suggesting that after a long period of relatively poor growth, those trees may have been bouncing back. Another statistical analysis would need to be done to properly analyze these trends (see Lloret et al. 2011 Oikos paper, and the R package pointRes [van der Maaten-Theunissen et al. 2015 Dendrochronologia] that implements the resistance/resilience/recovery metrics defined by Lloret et al).

Thank you. This is indeed an important addition. We added an appropriate test used the R package “QualInt” to implement the Gail Simon likelihood ratio test, which tests for differences in rank changes and provides p-values. The drop is significant ($p < 0.001$). The corresponding edits to the results and method sections read:

“Prior to this permanent rank change ($p < 0.001$, Gail-Simon Test), these more southern seed sources had shown equal productivity to the local seed sources.” (Lines 149-151)

“Rank changes of populations were tested according to Gail & Simon Likelihood Ratio Test⁶⁴ as implemented with the qualint function from the QualInt package⁶⁵” (Lines 253-255)

Comment 2.8: The most interesting result of the paper should be highlighted more strongly as a take-away message: there are opposing geographical clines of susceptibility to frost damage incurred by early-season frosts (“false spring”, frost ring position 2) vs. end-of-season frosts (blue rings and frost rings position 1), with northern provenances more susceptible to the former and southern provenances more susceptible to the latter. (By the way, the interpretation of this result is convincing and nicely argued.)

It is a good point and we now emphasize this more strongly:

“Using tree-ring signatures of cold damage in common garden trials designed to study genetic population differentiation, we find opposing geographic clines for spring frost and fall frost damage.” (Lines 5-7)

“Notably, the geographic cline in vulnerability to spring frosts (Fig 2c) appears opposite to geographic cline for fall frosts (Fig 2a and 2b).” (Lines 107-108)

“The study revealed opposing geographic clines for spring and fall frost damage, which both need to be considered to devise effective strategies to minimize risks in assisted migration prescription to address climate change.” (Lines 160-163)

Comment 2.10: Additionally, there is a parallel geographical cline with respect to absolute growth rate. These opposing geographic clines (particularly re: two kinds of frost damage) should be more strongly highlighted in the abstract (in the place of the sentence about rank-order change of growth of the southern-most populations) as a serious challenge to assisted migration strategies. I can only imagine that the negative evolutionary correlation uncovered by the authors in this study is the tip of the iceberg in terms of how assisted migration may fail (many other such negative evolutionary correlations are probably out there waiting to be discovered that constrain the success of assisted migration).

Thank you. As suggested, we now updated the abstract (see the last quotes in response to the above comment).

Reviewer #3 Comments

Comment 3.1: Montwe et al. analyse the effects of frost and extreme cold events at the beginning and the end of the vegetation period on growth and wood formation in trees. The authors make use of old provenance experiments which enable them to test for differences between 20 populations originating from major parts of the species (lodgepole pine) natural range and which allow for understanding effects on frost events under in situ conditions. They demonstrate that staining wood discs with a double staining procedure allowed identifying deformed tree rings and rings with incomplete lignified cells, and that these wood deformations can be clearly linked to frost events either at the beginning or the end of the vegetation period. The frequency of deformed rings differed significantly among populations and population groups and allowed to estimate effects of seed transfer on wood development and annual growth. Overall, the paper is well written, methodological sound and provides a fascinating new procedure to make use of tree provenance trials for climate change adaptation research. The methodology provided will certainly be adapted by forest scientist around the world for similar questions. Just be reading the paper, I found that the methodology might also support scientist to understand the risks associated with extended vegetation periods in climate change. The general topic of the paper is highly relevant: after about a decade of research and discussion of pro and cons of assisted migration in forest trees, assisted migration is now being integrated into reforestation and restoration practice. The given manuscript is highly needed as it sheds light on the potential drawback of artificial tree movement if populations are being moved too far from the climate conditions to which they are adapted to. This kind of risk assessment is required for seed transfer models. Thus, I see the paper as relevant and expect it to have a high impact on forest regeneration policies in northern boreal and temperate forests, where frost events will also in climate change be a significant driver of vegetation development. Give the high public and

political interest in future forest development, the paper should be considered as highlight paper in the New and Views of the Nature.

Overall, I highly recommend the publication of the given manuscript.

Thank you for your assessment.

Comment 3.2: Anyhow, there are a few minor issues that the authors should consider:

L131: "... northern populations are adapted to ..." instead of "must adapt"

Fixed as suggested: "...the northern population is adapted to a short growing season" (Line 130-131)

Comment 3.3: L141ff: I was surprised that the southern interior populations showed the highest fall frost damages, as these populations originate from the region where the three study sites are being located, so they should be locally well adapted. It was just when I checked tables S1 and S2 that I found that the test sites are located 300-800 m higher than the seed origins in southern interior. This certainly explains the higher late frost damages. Please, add few lines about the higher location of test sites to the discussion.

Thank you for pointing this out. This is indeed important information for proper interpretation of the results. We added a new panel to Figure 2 (h) to illustrate the climatic transfer distances, and we revised the corresponding result section:

"Another equally important finding is that it may not always be possible to transfer genotypes northward or upward in elevation as a management prescription to address climate change: Both the US and Southern Interior provenances were transferred to colder environments in this experiment (Fig 2h), with the test sites located either further north or higher in elevation than the provenance origins. Notably, the most productive Southern Interior provenances also showed the highest fall frost damage (Fig 2d-e, S1 a-b). Given that short growing seasons and low cumulative degree days are a likely cause of blue rings and frost rings in fall, a transfer of these seed sources to colder environments is likely to exacerbate this problem." (Line 139-146)

Comment 3.4: L223: what was interpolated? What interpolation technique was used?

The average (normal) climate for the period of 1961-1990 was derived with the software ClimateNA for the provenances' source locations. It is a method that uses the PRISM (Parameter-elevation Regressions on Independent Slopes Model) interpolation method (developed by Daly *et al.* 2008) to downscale observations from weather stations to provide estimates for any given latitude, longitude and elevation combination.

For more detail on the methodology used by this program to interpolate climate data, we refer to the accompanying publication for ClimateNA:

Wang, T., Hamann, A., Spittlehouse, D.L., and Carroll, C. 2016. Locally downscaled and spatially customizable climate data for historical and future periods for North America. PLoS One 11: e0156720.

Comment 3.5: Figures S1: its difficult to identify what the r in the Figure mean. I rather suggest a table clearly indicating which traits are being regressed to each other and giving the full stats of the regression (r, p-value, etc.)

Thank you for pointing out this out. We have added a more comprehensive description to the caption:

“Spearman correlation coefficients are found in the lower right corner of each panel. These correlations are between height or diameter and, depending on the panel, the three measures of cold (blue ring intensity, frost ring in position 1, frost ring in position 2)”

Comment 3.6: L236: here the full mixed model is being described. However, I cant see the full result. Please add another suppl. table with the complete stats for each of the tested effects.

Thank you. The p-values are reported in the text (Line 112-121). A table of multiple comparisons (Table S3) has also now been added that provides further statistical testing for pairwise comparisons among regions.

Reviewers' comments:

Reviewer #1 (Remarks to the Author):

The authors have nicely addressed my concerns and improved the manuscript where appropriate. I would like to thank them for the care with which they helped in clarifying some of my doubts/misunderstandings and how many of them have been also addressed by more specific text. I in particular really appreciated the improvement of figure 2 and the new figures add in the supplementary material.

Sincerely
Patrick Fonti

Reviewer #2 (Remarks to the Author):

Statistical analysis

1. The authors implemented a repeated measures permutational ANOVA to test for differences in tree ring damage intensity between the 4 regions of origin, though it's not clear to me whether this actually addressed the point about clustering in the data. Did this repeated measures permutational ANOVA include provenance, site, tree, and year random effects?

Out of curiosity then, given that a repeated measures term is included in the test of damage intensity, should this also be addressed in the test of damage occurrence?

2. I appreciate the figure showing climate variation in a reduced dimension (PCA) manner. I don't see anything really exciting or interesting popping out.

3. The issue, acknowledged by the authors, that small trees are more susceptible to the formation of frost rings in position 1 is not really addressed in the revision/rebuttal. The fact that Year is included as a random effect does not address the point. It would be so easy to include bole diameter (or radius...the cumulative sum of the rings) as a predictor in the GLMM. Then it would be possible to put a number on the effect of size vs. year-to-year climate variation on susceptibility to frost rings in position 1. Would it change the results, the big picture? No, so I don't think it merits rejecting the manuscript. It's more of a side issue that could have been handled better.

4. Regarding modeling the occurrence of damage as a function of climate in the GLMMs (rather than correlations). The authors argue that multiple regression would be difficult to interpret (too many climate predictors in one model) – this is not, in my opinion, a strong rebuttal. Multiple regression is a perfectly good and widely-used statistical tool across many scientific disciplines.

Dendrochronologists/climatologists have a long history of using single-variable correlations; the authors' response reflects that disciplinary inertia more than anything else. Certainly multiple regression could be used here in a hypothesis-testing framework – the authors have clear hypotheses about the mechanisms and conditions behind the three types of damage recorded in tree rings. Would this change the results? No, so I don't think it's a reason to reject the paper per se.

5. Regarding the rank change of growth of US populations following the spring 1992 frost event. There are still no error bars associated with the growth traces in Figure 2d-f. I don't know much about the Gail-Simon test, and not enough information is provided for me to understand why it is an appropriate test. A quick look at the documentation for the R package QualInt is not particularly revealing (<https://cran.r-project.org/web/packages/QualInt/QualInt.pdf>). I have a hard time believing the statement that the rank change is "permanent", given that the last data point for the US populations in Figure 2d-f shows its rank is #3 out of 4 regions, and its trend was upwards since 2002. At the very least, the word permanent should be dropped – what can be stated is that the rank changed within the time frame of the observations (which is limited to 2005).

Additional Points:

The revised title tries to say too much and is difficult to follow. Please consider a simplified and more general title. One possible suggestion:

"Cold adaptation recorded in tree rings highlights the risks associated with assisted migration"

In general, I think there is still room to frame this paper in a more general way – in terms of the problems associated with moving around locally-adapted genotypes. The authors have conceived of this paper in terms of "informing limits to seed transfer" of one species, but the results highlight a much more general problem, that fact that opposing geographic clines of local adaptation may make transfers both north and south fail – assisted migration might be a loose-loose strategy. Assisted migration is a solution that is proposed for more than just forests – it's proposed as a strategy for biodiversity in the face of climate change.

Small changes/corrections:

Line 28: a regular reforestation program...or regular reforestation programs (choose one or the other)

Line 28: change "This concept" to "Assisted migration"

Line 132: missing a word (the? a?) in the phrase "appears opposite to geographic cline for"

Line 156: The data suggest...(data are plural, not singular)

Lines 292-3: the description of the Gail-Simon test needs to be longer, with more information. Why this test?

Reviewer #3 (Remarks to the Author):

The authors have well addressed my comments on the first manuscript as well as the comments of the other referees. I do not have further comments on the contents and highly recommend its publication in Nature Communications.

Anyhow, here are a few minor issues on the wording:

L8: the authors use the phrase "false spring" in their abstract and also within further text. I strongly suggest to avoid this phrase in particular within the abstract because 1) "false spring" is not very clearly defined; and 2) the effects that the authors discuss in relation to spring frosts are not necessarily limited to "false springs" but might be observed also in earlier springs if some late frost occur and 3) a shift to earlier spring development is today and in the future rather the normal state than the "false". If the authors want to use the term "false spring" they need to give a proper definition.

L22: "... as part of regular reforestation programs."

L33-34: "Seeds collected from across a species range (i.e., provenances)...". This sounds like a mistakable definition of provenances... someone could take a seed collection across the species range as single provenance. I suggest either to remove the brackets and the explanation therein or to give a more precise definition of what a provenance is.

Best regards
S.Schueler

Please find our replies to specific reviewer comments inserted below in blue.

We thank all referees for agreeing to review the manuscript again, for their attentive eye and feedback. Their suggestions and recommendations significantly improved the manuscript, and our gratitude is expressed in the manuscript acknowledgements.

Reviewer #1 Comments:

Reviewer #1 (Remarks to the Author):

The authors have nicely addressed my concerns and improved the manuscript where appropriate. I would like to thank them for the care with which they helped in clarifying some of my doubts/misunderstandings and how many of them have been also addressed by more specific text. I in particular really appreciated the improvement of figure 2 and the new figures add in the supplementary material.

Sincerely
Patrick Fonti

Thank you for your positive assessment.

Reviewer #2 Comments:

Reviewer #2 (Remarks to the Author):

Thank you for your additional feedback. Please see below for descriptions of how we have addressed your concerns.

Statistical analysis

1. The authors implemented a repeated measures permutational ANOVA to test for differences in tree ring damage intensity between the 4 regions of origin, though it's not clear to me whether this actually addressed the point about clustering in the data. Did this repeated measures permutational ANOVA include provenance, site, tree, and year random effects? Out of curiosity then, given that a repeated measures term is included in the test of damage intensity, should this also be addressed in the test of damage occurrence?

We have revised the statistical analysis implementing the suggestion from your previous review. This involves testing group differences and the relationships with climate in one model. To implement this, we used a cumulative link mixed model that can handle the ordinal multinomial nature of the intensity score. This also allows to simplify the analysis because occurrence and intensity are now combined in one model, making the binomial glmm and the permutational

ANOVA redundant. While this makes the analysis more concise, we would like to emphasize that the different methods yield similar results.

While it is possible to account for repeated models in an ANOVA, mixed models are preferred when it comes to spatial or temporal correlation. We account for the repeated measures in the mixed model by including the random effect of tree id, thereby accounting for the correlation of the multiple observations per tree, following: Zuur, A. F. & Ieno, E. N. A protocol for conducting and presenting results of regression-type analyses. *Methods Ecol. Evol.* 7, 636–645 (2016).

2. I appreciate the figure showing climate variation in a reduced dimension (PCA) manner. I don't see anything really exciting or interesting popping out.

We now analyze the relationship with climate with covariates in the mixed model. Please find more detail below.

3. The issue, acknowledged by the authors, that small trees are more susceptible to the formation of frost rings in position 1 is not really addressed in the revision/rebuttal. The fact that Year is included as a random effect does not address the point. It would be so easy to include bole diameter (or radius...the cumulative sum of the rings) as a predictor in the GLMM. Then it would be possible to put a number on the effect of size vs. year-to-year climate variation on susceptibility to frost rings in position 1. Would it change the results, the big picture? No, so I don't think it merits rejecting the manuscript. It's more of a side issue that could have been handled better.

Testing the effect of stem diameter is now implemented with a mixed model. Diameter was found to be collinear with population. Therefore, we had to fit separate models for diameter and population. The results are now described in the text and model summaries were added to the results:

“Increasing diameter decreased the odds of observing more severe blue ring scores ($\chi^2=550.6$, $p<0.001$, Table S4), frost rings at position 1 ($\chi^2=398.9$, $p<0.001$, Table S5), and position 2 ($\chi^2=54.8$, $p<0.001$, Table S7).” Lines 88-90.

4. Regarding modeling the occurrence of damage as a function of climate in the GLMMs (rather than correlations). The authors argue that multiple regression would be difficult to interpret (too many climate predictors in one model) – this is not, in my opinion, a strong rebuttal. Multiple regression is a perfectly good and widely-used statistical tool across many scientific disciplines. Dendrochronologists/climatologists have a long history of using single-variable correlations; the authors' response reflects that disciplinary inertia more than anything else. Certainly multiple regression could be used here in a hypothesis-testing framework – the authors have clear hypotheses about the mechanisms and conditions behind the three types of damage recorded in tree rings. Would this change the results? No, so I don't think it's a reason to reject the paper per se.

We took your advice and included climate as covariates in the mixed model. To implement this efficiently, a cumulative link logistic mixed model was used (see above). We checked climate variables for collinearity and dropped variables with strong collinearity from the analysis. The results are now described in the text and model summary tables were added to the supplementary information. In addition, the description of the statistical analyses has been updated to reflect the improved statistical approach:

“Due to the ordinal scale of the blue and frost ring scores, we used a cumulative link mixed model, implemented in the `clmm` function of the `ordinal` package for R⁶⁰. This model was used to analyze differences between populations and to assess relationships with climate. A logit link and flexible thresholds between the ordinal scores was used. Site, block, provenance and tree identifier were specified as random effect with random intercepts. The unique tree identifiers were included to account for repeated measures taken on each tree⁶¹. Population was specified as fixed effect. In addition, several different climate variables were included as fixed effects. Continuous variables were scaled to ensure model convergence and tested for collinearity with Pearson's correlation coefficient. In case of collinearity between two variables, we chose the one deemed more biologically meaningful. Categorical variables were assessed for collinearity with boxplots⁶¹. Because stem diameter was found to be collinear with population, we tested the effect of diameter in separate models. Akaike's information criterion was used to select models with good fit and to simplify models. Effects were tested for significance using likelihood ratio tests.”
Lines 247-260.

5. Regarding the rank change of growth of US populations following the spring 1992 frost event. There are still no error bars associated with the growth traces in Figure 2d-f. I don't know much about the Gail-Simon test, and not enough information is provided for me to understand why it is an appropriate test. A quick look at the documentation for the R package `QualInt` is not particularly revealing (<https://cran.r-project.org/web/packages/QualInt/QualInt.pdf>). I have a hard time believing the statement that the rank change is “permanent”, given that the last data point for the US populations in Figure 2d-f shows its rank is #3 out of 4 regions, and its trend was upwards since 2002. At the very least, the word permanent should be dropped – what can be stated is that the rank changed within the time frame of the observations (which is limited to 2005).

We have now added error bars to Figures 2d-f and agree that they improve the figures. We have also removed both occurrences of the word “permanent” when discussing the rank change.

The Gail Simon likelihood ratio test is commonly used to assess the significance of rank changes (see, e.g. Isaac-Renton *et al.* 2014, *Global Change Biology*). We also refer readers to the original citation by Gail and Simon from 1985 (references 64) which describes the test in detail, although we provide additional information on how we apply it to our specific case:

“Rank changes of populations were tested according to the Gail & Simon Likelihood Ratio Test⁶⁴ as implemented with the `qualint` function from the `QualInt` package⁶⁵. The Gail Simon likelihood ratio test is a method for testing the significance of qualitative or crossover interactions. A qualitative interaction exists where levels of a group perform better under some circumstances

or treatment levels and vice versa. The null hypothesis is that the treatment effects in all sub-groups are in the same direction. In our case, we apply the Gail Simon test to evaluate the significance of rank changes in tree population growth under different time periods (pre- and post-frost in 1992).” Lines 270 -277.

Additional Points:

The revised title tries to say too much and is difficult to follow. Please consider a simplified and more general title. One possible suggestion:

“Cold adaptation recorded in tree rings highlights the risks associated with assisted migration”

This is a good title, thank you. We adopt your suggestion, with a minor modification:

“Cold adaptation recorded in tree rings highlights risks associated with climate change and assisted migration”

We added “climate change” because opposite clines indicate that there are risks to both climate change and assisted migration (discussed more below).

In general, I think there is still room to frame this paper in a more general way – in terms of the problems associated with moving around locally-adapted genotypes. The authors have conceived of this paper in terms of “informing limits to seed transfer” of one species, but the results highlight a much more general problem, that fact that opposing geographic clines of local adaptation may make transfers both north and south fail – assisted migration might be a loose-loose strategy. Assisted migration is a solution that is proposed for more than just forests – it’s proposed as a strategy for biodiversity in the face of climate change.

Thank you for this suggestion. We rephrased the introduction to highlight the more general problem of assisted migration.

“The pace of observed climate warming, particularly in northern latitudes, implies that environmental conditions are changing faster than plant populations can adapt, acclimate or migrate¹⁻³. This mis-match is expected to destabilize ecosystems by decreasing productivity while increasing mortality and susceptibility to insect attacks and diseases⁴⁻⁸. Maladapted plant populations may further contribute to climate change through reduced carbon sequestration, or may even turn into significant sources of carbon dioxide⁹. To mitigate such consequences, assisted migration has been proposed as part of regular reforestation programs^{10,11} and as part of conservation efforts¹². Assisted migration involves changing the genetic composition of plant populations by moving seed material to climate regions where they are anticipated to be well adapted in the future^{2,3,10,11}. For example, transferring seeds from southern, drought-tolerant tree populations may help more northern forests adapt to drier, warmer conditions^{2,3,13,14}. However, there are concerns that assisted migration may not be successful and lead to unintended consequences^{15,16}. For example, planting warm-adapted populations in anticipation of a warmer future could expose these seedlings to frost damage and compromise their

survival¹⁷⁻¹⁹. As climates warm and changing phenology interacts with temperature variability²⁰⁻²³, frost damage may remain a risk in decades to come. Understanding the adaptation of plant populations to cold – as well as to heat and drought – is critical for minimizing risks under changing climates and assisted migration²⁴.” Lines 16– 33.

We find direct evidence that assisted migration would pose a higher risk of cold damage to genotypes transferred to temporarily cooler environments, which is important for indicating limits of suitable seed transfer. The risks of inaction are not insignificant, either, however: The southward transfers tested here highlight the risks of high-magnitude warming for the northern population (assisted migration would not involve moving seeds to warmer climates).

Thus, as you point out, the opposite clines indicate that there are risks associated with both inaction (status quo reforestation) versus risks of assisted migration. Likely, the most suitable course of action would involve shorter seed transfer distances - equivalent to planning for a shorter time-frame with climate change (e.g. 20 years from now rather than a full rotation length).

It is also true that assisted migration is being considered more broadly in other widespread species where local adaptation exists. In our opening sentences, we now refer more generally to plants before indicating that our study focusses on tree populations. The introduction also speaks of assisted migration in general terms before using forests as an example. That said, we agree that the conclusions could be framed differently for broader impact. We have now added the following paragraph at the end of the discussion:

“Although maladaptation under climate change is of concern for forests due to long tree lifespan, genetic maladaptation may occur in other widespread species where local adaptation exists. Assisted migration is being considered more broadly as a means of conserving biodiversity under climate change⁵². If the opposite clines in cold adaptation found here apply more generally, this would highlight the risk trade-offs associated with inaction versus those of assisted migration. The most suitable course of action in forestry as well as conservation may thus involve shorter seed transfer distances, equivalent to planning for a shorter climate change time-frame.” Lines 171 – 177.

Small changes/corrections:

Line 28: a regular reforestation program...or regular reforestation programs (choose one or the other)

Thank you, good catch. Corrected.

Line 28: change “This concept” to “Assisted migration”

Changed.

Line 132: missing a word (the? a?) in the phrase “appears opposite to geographic cline for”

Good eye. We have added the word “the”.

Line 156: The data suggest...(data are plural, not singular)

Another good catch, thank you.

Lines 292-3: the description of the Gail-Simon test needs to be longer, with more information. Why this test?

We have now added additional information (please see response to comment number 5 above).

Reviewer #3 Comments:

Reviewer #3 (Remarks to the Author):

The authors have well addressed my comments on the first manuscript as well as the comments of the other referees. I do not have further comments on the contents and highly recommend its publication in Nature Communications.

Thank you for your positive assessment.

Anyway, here are a few minor issues on the wording:

L8: the authors use the phrase “false spring” in their abstract and also within further text. I strongly suggest to avoid this phrase in particular within the abstract because 1) “false spring” is not very clearly defined; and 2) the effects that the authors discuss in relation to spring frosts are not necessarily limited to “false springs” but might be observed also in earlier springs if some late frost occur and 3) a shift to earlier spring development is today and in the future rather the normal state than the “false”. If the authors want to use the term “false spring” they need to give a proper definition.

Good point. We removed the term from the abstract and define it in the results:

“Provenances from northern regions are sensitive to spring frosts, while the more productive provenances from central and southern regions are more susceptible to fall frosts.” Lines 7 – 9.

“As an example, the year 1992 had a high proportion of frost rings in position 2, and a false spring: an unusually early warm spell in April before day 100 (Fig 1f).” Lines 109 – 111.

L22: “... as part of regular reforestation programs.”

Good catch. Corrected.

L33-34: ”Seeds collected from across a species range (i.e., provenances)...”. This sounds like a mistakable definition of provenances... someone could take a seed collection across the species

range as single provenance. I suggest either to remove the brackets and the explanation therein or to give a more precise definition of what a provenance is.

Best regards
S.Schueler

Thank you, we agree that this could have caused misunderstanding. We now revise this with a more precise explanation:

“Seeds collected from across a species range (i.e., provenances are the geographic locations from which seed sources originate) are grown at multiple planting sites to reveal intra-specific genetic differentiation.” Lines 37 – 39.

REVIEWERS' COMMENTS:

Reviewer #2 (Remarks to the Author):

I want to re-emphasize that I feel that the manuscript reports original, novel results that are of interest for a broad global change and biodiversity science community, since assisted migration is a much-discussed strategy to mitigate the negative effects of climate change. This manuscript reports very interesting results that call into question how easy or successful assisted migration may be, i.e., because of opposing geographic adaptive clines.

I am satisfied with the revised statistical analysis. It has addressed my concerns about clustering in the data, pseudoreplication, repeated measures, etc. A single, combined analysis with a multinomial (ordinal) response is a much cleaner analysis. My remaining comments are a mixture of suggestions to further smooth the writing, a few corrections, and some thoughts about the bigger picture.

Line 3: "could be exacerbated by poleward planting of warm-adapted seed sources."

Line 6: "Using tree-ring signatures of cold damage from common garden trials..."

Lines 29-31: I feel this sentence is missing a reference to the literature predicting (and emerging evidence that it is happening) that climate change will lead to greater climate extremes. In particular, we are seeing slower jet stream behavior in the northern hemisphere, and increased sinuosity, with the result that we see phenomena such as "polar vortex" and "bomb cyclone" and "ridiculously resistant ridge"...i.e., not just droughts and pluvials, but also extreme cold temperatures. It would be highly relevant to this particular research to point out this connection.

Line 46: "genetic adaptation to cold in tree populations has been studied by observing tissue damage..."

Line 50: "Tree rings can provide..."

Line 61: "therefore act as long-term archives of cold damage."

Line 87: "Higher occurrences of blue and frost rings in the first ten years of the experiment..."

Lines 11-115: These sentences seem to belong to the following section on "genetic differences in susceptibility to cold damage"...they should be moved below. This consolidation would allow you to remove the redundancy between the sentence in lines 113-115 and the same information repeated in lines 124-126 about susceptibility of N populations to frost rings in position 2.

Line 129: susceptibility should be susceptible

Line 133-134: "is that northern populations are adapted..."

Line 136: "advantage of a limited growing season"

Line 139: the phrase "may be associated with forest health issues" is vague, and no citations are provided...can you reuse some of the citations from your first paragraph, if they are relevant?

Line 142: writing is awkward – please find a less wordy, more concise, elegant way to get your meaning across.

Lines 162-164: In my opinion, this result is of technical interest to a more constrained audience (a methodological result of interest to the dendro community) and does not rise to the level of interest for a broad audience that the other results do (i.e., opposing geographic clines, risks associated with assisted migration).

Line 171: "is of concern for forests due to the long lifespan of trees,"

Lines 188-190: "These 20 provenances were replicated at three experimental sites" "We chose these sites to have some replication while also representing an area..." Note that three replicates (common gardens) in the same region are, to some degree, spatially autocorrelated with one another and thus not "independent samples". And the use of the word "enough" is totally arbitrary...enough with respect to what? What analysis of statistical power or variability in the data? Better to remove these unsubstantiated, mildly incorrect word choices ("enough replications of independent samples")...which made for awkward writing as well.

Line 251: "were specified as random effects modifying the intercept(s?)."

Line 256: What does "more biologically meaningful" mean? This needs to be substantiated or explained. Be clear about the hypotheses being tested when certain climate predictors are included.

Line 258: refer to supplementary tables 3-8

Figure 2h: When I look at the equation for the x-axis of this figure (common garden deg C minus provenance deg C), I expect the values to be opposite in sign of what I see in the figure. The most northern provenances would have lower mean annual temperature (MAT) than the common garden locations, thus the MAT "residual" for a northern provenance should be positive not negative. Is it possible either that the equation is wrong or the scale on the axis is wrong???

Finally, the authors suggest that the solution to the challenges posed by complex adaptive clines is to limit the distance of assisted migration. What about recombining genotypes/phenotypes? I.E., breeding "super trees" that have both heat/drought AND cold tolerance? Either via traditional breeding or genomic approaches... There certainly are a number of people out there making their living working on tree genetics (breeding) and genomics.

Please find our replies to specific reviewer comments inserted below in blue.

We thank the referee for agreeing to review the manuscript again, for the attentive eye and feedback. The suggestions and recommendations significantly improved the manuscript.

Reviewer #2 Comments:

Reviewer #2 (Remarks to the Author):

I want to re-emphasize that I feel that the manuscript reports original, novel results that are of interest for a broad global change and biodiversity science community, since assisted migration is a much-discussed strategy to mitigate the negative effects of climate change. This manuscript reports very interesting results that call into question how easy or successful assisted migration may be, i.e., because of opposing geographic adaptive clines.

I am satisfied with the revised statistical analysis. It has addressed my concerns about clustering in the data, pseudoreplication, repeated measures, etc. A single, combined analysis with a multinomial (ordinal) response is a much cleaner analysis. My remaining comments are a mixture of suggestions to further smooth the writing, a few corrections, and some thoughts about the bigger picture.

Thank you for your assessment.

1. Line 3: “could be exacerbated by poleward planting of warm-adapted seed sources.”

Thank you. Changed as suggested.

2. Line 6: “Using tree-ring signatures of cold damage from common garden trials...”

Changed as suggested.

3. Lines 29-31: I feel this sentence is missing a reference to the literature predicting (and emerging evidence that it is happening) that climate change will lead to greater climate extremes. In particular, we are seeing slower jet stream behavior in the northern hemisphere, and increased sinuosity, with the result that we see phenomena such as “polar vortex” and “bomb cyclone” and “ridiculously resistant ridge”...i.e., not just droughts and pluvials, but also extreme cold temperatures. It would be highly relevant to this particular research to point out this connection.

Thank you. To reflect this suggestion, we have now added two references about increased likelihood of extremes due to climate change, including extreme cold. These references cite Cohen *et al.* 2018 (Nature Communications) and Kretschmer *et al.* 2018 (American Meteorological Society). The revised sentence has shifted from “may remain a risk” to “will remain a risk”, and reads:

“As climates warm and changing phenology interacts with increasing temperature variability²⁰⁻²⁵, frost damage will remain a risk in the future.” Lines 29-31

4. Line 46: “genetic adaptation to cold in tree populations has been studied by observing tissue damage...”

Thank you. Changed as suggested.

5. Line 50: “Tree rings can provide...”

Changed as suggested.

6. Line 61: “therefore act as long-term archives of cold damage.”

Changed as suggested.

7. Line 87: “Higher occurrences of blue and frost rings in the first ten years of the experiment...”

Changed as suggested.

8. Lines 11-115: These sentences seem to belong to the following section on “genetic differences in susceptibility to cold damage”...they should be moved below. This consolidation would allow you to remove the redundancy between the sentence in lines 113-115 and the same information repeated in lines 124-126 about susceptibility of N populations to frost rings in position 2.

The sentences were moved as suggested and the text consolidated. The revised sentences read:

“In contrast, Northern provenances were the most susceptible to spring frost damage as indicated by frost rings in position 2 ($\chi^2= 9.2$, $p=0.026$, Supplementary Table 6). Notably, the geographic cline in vulnerability to spring frosts (Fig 2c) appears opposite to the geographic cline for fall frosts (Fig 2a and 2b), with the most northern and coldest location of origin showing the highest intensity of frost rings in position 2 (Fig 2g and 2h). Similarly, correlations to climate of seed origin (Table 2) confirm that provenances adapted to northern, colder environments with larger temperature differences and shorter growing seasons incur less blue rings and fall frost damage but are more susceptible to spring frosts.” Lines 131-139

9. Line 129: susceptibility should be susceptible

Changed. Thanks for this catch.

10. Line 133-134: “is that northern populations are adapted...”

Changed as suggested.

11. Line 136: “advantage of a limited growing season”

Changed as suggested.

12. Line 139: the phrase “may be associated with forest health issues” is vague, and no citations are provided...can you reuse some of the citations from your first paragraph, if they are relevant?

This is true, thank you. The revised sentence reads:

“Moreover, the resulting frost damage can reduce growth and survival¹⁹. Such damage could also reduce wood quality and value by creating weaknesses and defects in the timber⁵³.” Lines 149-151

13. Line 142: writing is awkward – please find a less wordy, more concise, elegant way to get your meaning across.

The shortened sentence reads:

“Another equally important finding is that this study confirms the need for transfer limits under assisted migration.” Lines 153-155

14. Lines 162-164: In my opinion, this result is of technical interest to a more constrained audience (a methodological result of interest to the dendro community) and does not rise to the level of interest for a broad audience that the other results do (i.e., opposing geographic clines, risks associated with assisted migration).

Removed for conciseness.

15. Line 171: “is of concern for forests due to the long lifespan of trees,”

Thank you. Changed as suggested.

16. Lines 188-190: “These 20 provenances were replicated at three experimental sites” “We chose these sites to have some replication while also representing an area...” Note that three replicates (common gardens) in the same region are, to some degree, spatially autocorrelated with one another and thus not “independent samples”. And the use of the word “enough” is totally arbitrary...enough with respect to what? What analysis of statistical power or variability in the data? Better to remove these unsubstantiated, mildly incorrect word choices (“enough replications of independent samples”)...which made for awkward writing as well.

This is a good suggestion, we removed the sentence.

17. Line 251: “were specified as random effects modifying the intercept(s?).”

Rephrased as suggested:

“Site, block, provenance and tree identifier were specified as random effects modifying the intercepts.” Lines 266-268

18. Line 256: What does “more biologically meaningful” mean? This needs to be substantiated or explained. Be clear about the hypotheses being tested when certain climate predictors are included.

Thank you, we made the suggested change and rephrased the sentences to also remove redundancies with the previous paragraph.

“We derived daily temperature minimums and averages from the ECMWF’s ERA interim data for 1979 to 2006⁶¹ for the grid-points closest to our site coordinates. This data is available through KNMI’s Climate Explorer (climexp.knmi.nl/). To capture general temperature conditions, as well as cold and frost events on an annual level, we calculated several additional climate variables. The start of the growing season was set as the last day in spring on which minimum temperatures dropped below 0 °C (Julian date). The end of the growing season was set to the Julian day on which the first frost (<0 °C) occurred again in autumn. Growing season length was calculated as the number of Julian days between the start and the end of the growing season. Growing degree days were calculated to as degree days above 5°C to characterize conditions within the growing season. To capture spells of warm weather early in the spring, before the last frosts have occurred, the Julian day of the year at which the sum of growing degree days over 5°C exceeded 100 was calculated. To investigate the relationship between blue and frost rings and the climate at the seed source locations, we interpolated climate normal data for the 1961-1990 period with the software ClimateNA⁶². This period was chosen because of wide spatial representation of climate stations and because it precedes the most recent anthropogenic climate warming signal.” Lines 240-259

“Climate variables were selected based on hypotheses about the cause of blue and frost rings. For blue rings, the Julian dates of the start and end of the growing season as well as the annual sum of growing degree days over 5 °C were tested. For frost rings at position 1, the Julian date of the end of the growing season and the annual sum of growing degree days over 5°C of the previous year were included. Finally, for frost rings at position 2, the Julian date on which a sum of degree days over 5°C exceeded 100 and the Julian date of the start of the growing season was tested. Categorical variables were assessed for collinearity with boxplots⁶³.” Lines 273-280

19. Line 258: refer to supplementary tables 3-8

Done. Thank you.

“Because stem diameter was found to be collinear with population, we tested the effect of diameter in separate models (Supplementary Tables 3-8).” Lines 280-282

20. Figure 2h: When I look at the equation for the x-axis of this figure (common garden deg C minus provenance deg C), I expect the values to be opposite in sign of what I see in the figure. The most northern provenances would have lower mean annual temperature (MAT) than the common garden locations, thus the MAT “residual” for a northern provenance should be positive not negative. Is it possible either that the equation is wrong or the scale on the axis is wrong???

Thank you for catching this mistake. Indeed, the equation should be Provenance °C – Site °C. Fixed in the figure.

21. Finally, the authors suggest that the solution to the challenges posed by complex adaptive clines is to limit the distance of assisted migration. What about recombining genotypes/phenotypes? I.E., breeding "super trees" that have both heat/drought AND cold tolerance? Either via traditional breeding or genomic approaches... There certainly are a number of people out there making their living working on tree genetics (breeding) and genomics.

A sentence and citation to this effect has now been added to the discussion:

“In tree improvement programs, breeding populations could be screened for cold tolerance^{31,54}.” Lines 182-183